# Geo-R1: Improving Few-Shot Geospatial Referring Expression Understanding with Reinforcement Fine-Tuning

## Abstract

Referring expression understanding in remote sensing poses unique challenges, as it requires reasoning over complex object–context relationships. While supervised fine-tuning (SFT) on multimodal large language models achieves strong performance with massive labeled datasets, they struggle in data-scarce scenarios, leading to poor generalization. To address this limitation, we propose Geo-R1, a reasoning-centric reinforcement fine-tuning (RFT) paradigm for few-shot geospatial referring. Geo-R1 enforces the model to first generate explicit, interpretable reasoning chains that decompose referring expressions, and then leverage these rationales to localize target objects. This "reason first, then act" process enables the model to make more effective use of limited annotations, enhances generalization, and provides interpretability. We validate Geo-R1 on three carefully designed few-shot geospatial referring benchmarks, where our model consistently and substantially outperforms SFT baselines. It also demonstrates strong cross-dataset generalization, highlighting its robustness. Code and data will be released at `http://geo-r1.github.io`.

## 1 Introduction

Vision language models (VLMs) have become a critical tool for remote sensing imagery (RSI) understanding (Li et al., 2024d; Weng et al., 2025). By coupling natural language with RSI, VLMs can drive a wide spectrum of tasks in the RS domain, such as image captioning, visual question answering, referring expression comprehension (REC), referring expression segmentation (RES) (Li et al., 2024d; Zhou et al., 2024a). Among these capabilities, REC and RES tasks are especially important: both require the model to resolve free-form linguistic descriptions (e.g., "a small vehicle is situated at the bottom right adjacent to a large vehicle") into concrete, spatially localized predictions (bounding boxes or segmentation masks) in high-resolution aerial images. We henceforth use the term *Referring Expression Understanding* (REU) to denote a unified framework encompassing both REC and RES, where the task is to take an image and a text query as input and output one or more target objects.

Although recent works (Kuckreja et al., 2024; Yuan et al., 2024; Zhou et al., 2024b) have achieved remarkable progress on REU tasks with supervised finetuning (SFT), these methods are highly dependent on large-scale training labels. High-quality REU supervision demands not only image-level labels but also precise language–region alignment at the object and region levels. Creating such associations in overhead imagery requires expertise and careful tooling: annotators must parse complex scene layouts, disambiguate visually similar man-made structures, and write unambiguous referring expressions before drawing spatially accurate boxes or masks. Compared with image-level labels, these fine-grained annotations are orders of magnitude more labor-intensive. For example, VRSBench (Li et al., 2024c) costs 1,004 labor hours for label verification only.

This reality makes few-shot learning (e.g., only 10 samples are provided for each category) in REU valuable. Previous works, such as RS-CLIP(Li et al., 2023) and RemoteCLIP (Liu et al., 2024a) have demonstrated that finetuning CLIP (Radford et al., 2021) on a few samples can yield strong results for scene classification. However, these advances cannot be directly carried over to REU since region-level grounding is harder than scene-level classification. Moreover, object relations

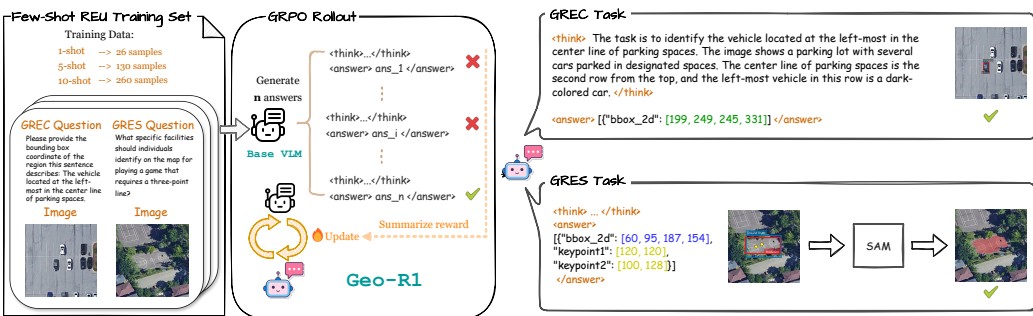

Figure 1: Geo-R1 method overview. Geo-R1 is trained on a few labeled samples with reinforcement learning (e.g., GRPO (Shao et al., 2024)) and can identify target objects (bounding boxes or masks) from an input image and text query while providing the reasoning process.

are complex for REU, requiring relational reasoning and disambiguation among visually similar structures. This raises the question: *with only a handful of aligned examples for each category, can a VLM learn to accurately ground language in remote sensing images?*

Driven by the impressive reasoning capabilities of OpenAI o1 (Jaech et al., 2024) and DeepSeek-R1 (Guo et al., 2025), reinforcement learning (RL) has become a powerful post-training paradigm for augmenting the reasoning capabilities of LLMs during post-training. RL explicitly encourages intermediate "thinking" steps, and forces the model learns to reason before committing to a prediction. This reasoning-first behavior is particularly well suited to few-shot REU: reasoning steps (e.g., "My intuition leads me to identify the vehicle sitting in the circular opening near the roadway as the small vehicle.") serve as a transferable experience that generalizes better across different text-image samples than directly outputting a box/mask from next-token-prediction supervision.

In this work, we introduce a reasoning-centric RL post-training method, Geo-R1, which leverages task-specific reward functions to address few-shot REU. Geo-R1 encourages the model to generate explicit reasoning—intermediate hypotheses that parse the referring expression, identify contextual anchors, and iteratively refine localization—thereby regularizing learning and improving generalization. Unlike SFT, which relies on a single teacher-forced trajectory with a differentiable surrogate loss, Geo-R1 explores multiple reasoning chains and proposals, extracting advantages from $N$-way comparisons to provide denser and richer supervision per example, making better use of few-shot samples. Moreover, for RES, Geo-R1 directly optimizes a task-aligned *MaskGIoU* reward through the non-differentiable "BBox + SAM" pipeline (Ravi et al., 2025), enabling end-to-end training for dense prediction—a capability infeasible under SFT. Method overview can be found in Fig. 1.

In our experiments, we observe three consistent advantages from RL over SFT baselines for few-shot REU in remote sensing images. (1) With the same small number of labeled examples, our RFT-based reasoning model substantially outperforms SFT-based models on few-shot REU tasks. (2) In cross-dataset evaluation, our RFT-based model remarkably outperforms SFT counterparts, suggesting the reasoning model has stronger cross-dataset generalization than non-reasoning models. (3) The learned reasoning traces are useful and reasonable, utilizing the spatial and semantic cues that benefit the final localization, which provides a great interpretability. We further establish three few-shot benchmarks and define a few-shot protocol for REU. In summary, our contributions are listed below:

- To the best of our knowledge, we are the first to explore Referring Expression Understanding (REU) for aerial image understanding under few-shot settings. To facilitate rigorous and reproducible evaluation, we create VRSBench-FS, EarthReason-FS, and NWPU-FS, establishing standardized protocols for few-shot REU in remote sensing.

- We define task-aligned rewards and a reasoning-centric RL recipe, including BBoxIoU reward for REC and a MaskGIoU reward for RES. We introduce the RL-trained reasoning models (Geo-R1) that generate concise grounding rationales for these tasks.

- Across all three benchmarks, our Geo-R1 models consistently outperform SFT under identical few-shot budgets, while exhibiting stronger generalization across datasets and providing human-auditable reasoning traces that explain successes and failures.

## 2 TASK AND METHODOLOGY

This section details the adaptation of the GRPO algorithm from language-only tasks to vision-language tasks. Then, we introduce and formally define the REU task under few-shot settings. Finally, we discuss how to apply GRPO to these tasks with customized task-specific reward functions.

### 2.1 GRPO: FROM LLM TO VLM

Group Relative Policy Optimization (GRPO) (Shao et al., 2024) is a reinforcement learning framework that removes the dependence on a value model and instead utilizes rule-based reward functions. The GRPO algorithm begins by sampling $N$ candidate outputs $\{o_1, \ldots, o_N\}$ from the current policy model $\pi_\theta$ for a given query prompt $q$. Each response $o_i$ is then evaluated by a reward function $R(q, o_i)$ to obtain a raw reward score $r_i$. To measure the relative quality of each response within the sampled group, GRPO standardizes the raw rewards to obtain the advantage value, as shown in Eq. 1. The advantage value $\hat{A}_i$ denotes the normalized advantage of the response $o_i$ relative to other samples within the group.

$$\hat{A}_i = \frac{r_i - \text{mean}\{r_1, r_2, \ldots, r_N\}}{\text{std}\{r_1, r_2, \ldots, r_N\}} \tag{1}$$

The policy $\pi_\theta$ is updated with a training objective (Eq. 2), designed to encourage the generation of responses with higher advantages.

$$\mathcal{J}_{\text{GRPO}}(\theta) = \mathbb{E}_{\{o_i\}_{i=1}^N \sim \pi_{\theta_{\text{old}}}(\cdot|q)} \left[ \frac{1}{N} \sum_{i=1}^N \left( \min\left( c_1 \cdot \hat{A}_i, \ c_2 \cdot \hat{A}_i \right) - \beta D_{\text{KL}}(\pi_\theta || \pi_{\text{ref}}) \right) \right], \tag{2}$$

where

$$c_1 = \frac{\pi_\theta(o_i \mid q)}{\pi_{\theta_{\text{old}}}(o_i \mid q)}, c_2 = \text{clip}\left( \frac{\pi_\theta(o_i \mid q)}{\pi_{\theta_{\text{old}}}(o_i \mid q)}, 1 - \varepsilon, 1 + \varepsilon \right). \tag{3}$$

Here, $D_{\text{KL}}(\pi_\theta || \pi_{\text{ref}})$ denotes the KL divergence between the current policy $\pi_\theta$ and the reference policy $\pi_{ref}$, which serves as a regularization term to prevent large deviations. The clipping mechanism in $c_2$ stabilizes training by constraining the policy update ratio.

For LLMs on tasks with definitive answers, like mathematical reasoning, the reward can be calculated using a rule-based verifiable reward function. uilding on GRPO, DeepSeek-R1 (Guo et al., 2025) demonstrates that such rewards enable models to produce both final answers and coherent reasoning traces. This approach has been successfully extended to VLMs by converting visual metrics into tailored reward signals (Shen et al., 2025; Liu et al., 2025a;b).

### 2.2 FEW-SHOT REFERRING EXPRESSION UNDERSTANDING TASK

*We define referring expression understanding as a unified framework for object recognition (either detection/segmentation) from referring expressions.* Given an image $I$ and a textual query $q$, a vision–language model (VLM) $\mathcal{F}$ predicts one or more target objects, as formulated in Eq. 4:

$$\{O_1, \ldots, O_N\} = \mathcal{F}(I, q), \tag{4}$$

where each $O_i$ denotes a predicted object parsed from VLM text outputs, and $N$ denotes the number of parsed objects. We define REC, Visual Grounding (VG) (Plummer et al., 2015), and Open-Vocabulary Detection (OVD) as instances of *Generalized REC (GREC)*, where each referred object $O_i$ is represented by a bounding box. Likewise, we define RES and Open-Vocabulary Segmentation (OVS) (Wu et al., 2024) as instances of *Generalized RES (GRES)*, where each object $O_i$ is represented by an instance mask.

In this work, we focus on three representative REU tasks: (i) REC, which targets single-object detection from complex reasoning queries; (ii) OVD, which addresses multi-object detection from

class-based queries; and (iii) GRES, which requires multi-object segmentation from complex reasoning queries. All tasks are studied under few-shot settings. In our formulation, each shot label refers to a annotated bounding box or mask. Specifically, in the GREC setup, one "shot" is defined as an image–query–box triplet, while in GRES, one "shot" corresponds to an image–query–mask triplet. Importantly, a ground-truth mask may include multiple valid instances for a single query (Li et al., 2025b). Among these tasks, GRES is the most challenging, as it requires the model to generate accurate segmentation masks for (multiple) objects described by natural-language queries in aerial images (Yuan et al., 2024).

The few-shot setting substantially increases task difficulty by requiring models to generalize from only a handful of labeled examples, in contrast to large-scale datasets such as VRSBench (Li et al., 2024c) (36k training examples), and DIOR-RSVG (27k) (Zhan et al., 2023). Few-shot REU is particularly challenging due to: (1) *visual diversity*, arising from large variations in object size, orientation, appearance, and inter-object relationships; and (2) *description diversity*, as natural language queries may vary in structure, vocabulary, abstraction level, and reasoning complexity. These factors jointly make few-shot REU a more realistic yet significantly harder problem compared to conventional large-scale training scenarios.

## 2.3 REWARD DESIGN

Following DeepSeek-R1, the reward function of Geo-R1 includes a task-agnostic format reward and a task-specific metrics reward. The format reward is applied uniformly across all tasks, whereas the metric reward is selected according to the requirements of each specific task.

### 2.3.1 FORMAT REWARD

To ensure reliable parsing and evaluation, the model's output must follow a well-defined structure. We define a binary format reward that checks whether the response conforms to this structure. The output must be wrapped in reasoning tags `<think>...</think>` and `<answer>...</answer>`. The format reward is defined as:

$$R_{\text{format}}(q, o) = \begin{cases} 1, & \text{if output follows the expected format} \\ 0, & \text{otherwise} \end{cases} \tag{5}$$

### 2.3.2 METRICS REWARD

**GREC**. For the REC task, the VLM predicts a single object bounding box, i.e., $b_{\text{pred}} = \mathcal{F}(I, q)$. An IoU reward can be calculated by comparing $b_{\text{pred}}$ with the ground-truth box $b_{\text{gt}}$. For the OVD task, the VLM predicts is a set of box–label pairs, i.e., $\mathbb{B}_{\text{pred}} = \{(b_{\text{pred}}^i, c_{\text{pred}}^i)\}_{i=1}^N$, where $b_{\text{pred}}^i$ denotes predicted bounding box, $c_{\text{pred}}^i$ denotes category label. We then calculate reward as mAP[1] between $\mathbb{B}_{\text{pred}}$ and corresponding ground truth $\mathbb{B}_{\text{gt}}$, along with a penalty coefficient for overlength predictions. The metrics reward for GREC task is defined as follow:

$$R_{\text{metrics}}(q, o) = \begin{cases} \text{IoU}(b_{\text{pred}}, b_{\text{gt}}), & \text{for REC task} \\ \min\left(1, \sqrt{\frac{N_{gt}}{N}}\right) \cdot \text{mAP}(\mathbb{B}_{\text{pred}}, \mathbb{B}_{\text{gt}}), & \text{for OVD task} \end{cases} \tag{6}$$

where $N_{gt}$ denotes the number of ground truth objects.

**GRES**. For GRES task, the VLM model is prompted to output a set of box–point pairs, $\mathbb{B}_{\text{pred}} = \{(b_{\text{pred}}^i, p_{\text{pred}}^i)\}_{i=1}^N$, where $b_{\text{pred}}^i$ denotes a predicted bounding box and $p_{\text{pred}}^i$ denotes the associated keypoints. These predictions are then provided as prompts to a frozen SAM to generate final instance masks $\mathbb{M}_{\text{pred}}$. Each predicted instance mask is trimmed to ensure its boundary does not exceed that of the corresponding bounding box. Finally, all instance masks are combined by taking their union to form a single predicted segmentation. Given ground truth instance masks $\mathbb{M}_{\text{gt}}$, the metrics reward for GRES task is defined as follows:

$$R_{\text{metrics}}(q, o) = \text{MaskGIoU}(\mathbb{M}_{\text{pred}}, \mathbb{M}_{\text{gt}}). \tag{7}$$

We follow LISA (Lai et al., 2024) to calculate mask GIoU.

---

[1]We set the confidence score of all predicted bounding boxes to 1.

## 3 MAIN EXPERIMENT

### 3.1 EXPERIMENT SETUP

**Datasets.** Unlike conventional few-shot learning (e.g., Prototypical Networks (Snell et al., 2017) and TFA (Wang et al., 2020)), we do not partition the dataset into base and novel classes. Instead, we treat all classes as novel and provide only a few labeled examples per class. We construct instruction-following few-shot datasets for the GREC and GRES tasks by deriving them from the training sets of three widely used remote sensing benchmarks: VRSBench Li et al. (2024c), NWPU VHR-10 (Cheng et al., 2014), and EarthReason (Li et al., 2025b). Configurations and statistics are summarized in Table 1. *The term "shot" defines the number of samples per object category.* For the OVD task, we select four classes on which the baseline model (Qwen2.5-VL-3B) demonstrated decent performance. We select all categories from the training set for other tasks. The low-shot dataset is a subset of the high-shot dataset. To evaluate cross-dataset generalization, we further evaluate zero-shot performance on DIOR-RSVG (Zhan et al., 2023) and RRSIS-D (Yuan et al., 2024) datasets.

Table 1: Overview of our Few-Shot Referring Expression Understanding Datasets.

| Dataset Name | Source Dataset | Task | # Categories | # Shots | # VQAs | # Images | Shot Definition |
|---|---|---|---|---|---|---|---|
| VRSBench-FS | VRSBench | REC | 26 | {10, 5, 1} | 260 | 254 | image-query-box |
| NWPU-FS | NWPU VHR-10 | OVD | 4 | {10, 5} | 25 | 25 | image-query-box |
| EarthReason-FS | EarthReason | GRES | 24 | {10, 5, 1} | 240 | 240 | image-query-mask |

**Model and Training Details.** We adopt Qwen2.5-VL-3B-Instruct (Bai et al., 2025) as base model. Our implementation is built on the VLM-R1[2] and Easy-R1[3] codebase. Unless otherwise specified, we strictly inherit the default hyperparameters without manual tuning. We set the same batch size for different post-training paradigm. We trained comparing models for 30 epochs, with early stopping when the reward converged. All experiments are conducted on $8 \times$ H100 GPUs, and a full training run takes approximately 10 to 20 hours. Prompt templates are shown in Appendix C. We apply thinking prompts for RL-based Paradigms. We adopt GRPO as our primary RL-based post-training paradigm. For SFT-based post-training, we perform visual instruction tuning with standard next token prediction (NTP) loss, implemented via LLaMA-Factory (Zheng et al., 2024).

### 3.2 FEW-SHOT GENERALIZED REFERRING EXPRESSION COMPREHENSION - REC

**Task Evaluation.** Performance on the REC subtask is measured by Acc@$\tau$ (a prediction is correct if its box IoU with the ground truth exceeds $\tau$) in the test set of VRSBench. We report metrics for Acc@0.5 and Acc@0.7. The experiments are conducted in 1-shot, 5-shot, and 10-shot configurations, with "Unique," "Non-Unique," and overall results reported. This evaluation compares the SFT method against two RL-based approaches, GRPO (Shao et al., 2024) and DAPO (Yu et al., 2025). We highlight the performance gap in red.

**Results.** Table 2 compares models trained on the full VRSBench (Full Amount Fine-tune) against few-shot models (1/5/10-shot Fine-tune). The few-shot results include both SFT-based models and our RL-tuned models. Performance data for the full-data baselines (except Qwen2.5-VL) are taken from the original VRSBench paper. The results reveal a clear performance hierarchy: RL-based post-training methods consistently and significantly outperform the SFT approach across all settings and metrics. This advantage is substantial; for example, in the 10-shot overall setting, our GRPO-based model achieves an Acc@0.5 score 12.30% higher than its SFT counterpart. Remarkably, our 10-shot GRPO model using only 260 samples, 0.71% data, achieves a score that surpasses all evaluated models (except Qwen2.5-VL) trained on all 36,313 samples.

Within RL-based approaches, DAPO consistently outperforms GRPO across nearly all scenarios, indicating that more effective RL training could further enhance performance in few-shot settings. Moreover, the gains from RL-based methods are more pronounced on the Unique subset than on the Non-Unique subset, suggesting that RL approaches provide a larger boost on simpler tasks that do not require distinguishing between same-category distractors.

---

[2]https://github.com/om-ai-lab/VLM-R1
[3]https://github.com/hiyouga/EasyR1

Table 2: Performance on VRSBench for the REC task. We report grounding accuracy at IoU thresholds of 0.5 and 0.7. Unique and Non-Unique indicate whether a referred object is the only instance of its category in the image or not.

| Method | Base LLM | Unique | | Non-Unique | | Overall | |
|---|---|---|---|---|---|---|---|
| | | Acc@0.5 | Acc@0.7 | Acc@0.5 | Acc@0.7 | Acc@0.5 | Acc@0.7 |
| **Full Amount Fine-tune (36,313 samples)** | | | | | | | |
| LLaVA-1.5 (Liu et al., 2024b) | Vicuna1.5-7B | 51.10 | 16.40 | 34.80 | 11.50 | 41.60 | 13.60 |
| Mini-Gemini (Li et al., 2024f) | Gemma-7B | 41.10 | 9.60 | 22.30 | 4.90 | 30.10 | 6.80 |
| MiniGPT-v2 (Chen et al., 2023) | Vicuna1.5-7B | 40.70 | 18.90 | 32.40 | 15.20 | 35.80 | 16.80 |
| GeoChat (Kuckreja et al., 2024) | Vicuna1.5-7B | 57.40 | 22.60 | 44.50 | 18.00 | 49.80 | 19.90 |
| Qwen2.5-VL (Bai et al., 2025) | Qwen2.5-3B | 66.54 | 36.77 | 60.32 | 36.30 | 62.91 | 36.50 |
| **Zero-shot Baseline** | | | | | | | |
| GPT-4V (OpenAI, 2024) | GPT-4 | 8.60 | 2.20 | 2.50 | 0.40 | 5.10 | 1.10 |
| Qwen2.5-VL w/o thinking | Qwen2.5-3B | 43.10 | 25.10 | 33.46 | 18.01 | 37.48 | 20.97 |
| Qwen2.5-VL w/ thinking | Qwen2.5-3B | 46.18 | 26.90 | 35.22 | 18.87 | 39.79 | 22.22 |
| **1-shot Fine-tune (26 samples)** | | | | | | | |
| Qwen2.5-VL-SFT | Qwen2.5-3B | 34.32 | 18.87 | 31.62 | 16.35 | 32.75 | 17.40 |
| Geo-R1 (GRPO) | Qwen2.5-3B | **52.17** (+17.85) | 31.18 (+12.31) | 41.21 (+9.59) | 23.04 (+6.69) | 45.78 (+13.03) | 26.43 (+9.03) |
| Geo-R1 (DAPO) | Qwen2.5-3B | 51.72 (+17.40) | **31.68** (+12.81) | **42.13** (+10.51) | **24.50** (+8.15) | **46.13** (+13.38) | **27.50** (+10.10) |
| **5-shot Fine-tune (130 samples)** | | | | | | | |
| Qwen2.5-VL-SFT | Qwen2.5-3B | 36.98 | 16.61 | 33.94 | 17.17 | 35.21 | 16.94 |
| Geo-R1 (GRPO) | Qwen2.5-3B | 54.11 (+17.13) | 31.35 (+14.74) | 42.98 (+9.04) | 23.98 (+6.81) | 47.62 (+12.41) | 27.06 (+10.12) |
| Geo-R1 (DAPO) | Qwen2.5-3B | **55.73** (+18.75) | **32.19** (+15.58) | **44.19** (+10.25) | **24.86** (+7.69) | **49.00** (+13.79) | **27.92** (+10.98) |
| **10-shot Fine-tune (260 samples)** | | | | | | | |
| Qwen2.5-VL-SFT | Qwen2.5-3B | 41.81 | 18.59 | 35.78 | 17.20 | 38.29 | 17.78 |
| Geo-R1 (GRPO) | Qwen2.5-3B | 57.27 (+15.46) | 35.61 (+17.02) | 45.81 (+10.03) | 27.03 (+9.83) | 50.59 (+12.30) | 30.61 (+12.83) |
| Geo-R1 (DAPO) | Qwen2.5-3B | **59.49** (+17.68) | **37.11** (+18.52) | **47.91** (+12.13) | **28.07** (+10.87) | **52.74** (+14.45) | **31.84** (+14.06) |

### 3.2.1 FEW-SHOT GENERALIZED REFERRING EXPRESSION COMPREHENSION - OVD

**Task Evaluation.** For the OVD task, we evaluate performance using the COCO-style mean Average Precision (mAP) in the test set of NWPU VHR-10 (Cheng et al., 2014). Our evaluation compares the SFT method against our GRPO approach. Experiments are run in 5/10-shot settings. Results are reported for the following four categories: airplane (PL), ship (SH), ground track field (GTF), and vehicle (VH). We intentionally exclude the 1-shot setting because training on a single instance would bias the model toward predicting a single instance per image, creating an inconsistency between the training and testing sets.

**Results.** Table 3 presents the OVD performance of SFT and GRPO tuned models. A notable observation is that SFT can be detrimental with extremely limited data. In both 10-shot and 5-shot settings, SFT-based models fail to surpass the performance of the zero-shot baseline in three out of four categories (airplane, ship, and vehicle). This suggests that the limited training data lacks intra-class diversity, causing the model to memorize the specific and even spurious features of the few samples rather than the general concept of the class, leading to overfitting, degrading the model's detection ca-

Table 3: Performance on NWPU for the OVD task. We report mAP in COCO style.

| | PL | SH | GTF | VH | Avg. |
|---|---|---|---|---|---|
| **Zero-shot Baseline** | | | | | |
| Qwen2.5-VL w/o thinking | 23.79 | 25.34 | 44.13 | 24.04 | 29.33 |
| Qwen2.5-VL w/ thinking | 25.17 | 21.85 | 57.08 | 23.95 | 32.01 |
| **5-shot Fine-tune (20 samples)** | | | | | |
| Qwen2.5-VL-SFT | 6.32 | 22.33 | 65.48 | 12.36 | 26.62 |
| Geo-R1 (GRPO) | **21.74** | **25.42** | **70.23** | **15.40** | **33.20** |
| **10-shot Fine-tune (40 samples)** | | | | | |
| Qwen2.5-VL-SFT | 15.76 | 21.90 | 68.42 | 14.73 | 30.20 |
| Geo-R1 (GRPO) | **25.76** | **28.12** | **69.24** | **16.57** | **34.92** |

pabilities. In contrast, the GRPO-tuned model consistently outperforms the SFT model across all categories and settings, demonstrating that RL is more efficient for learning OVD from a few examples. More importantly, the advantage of GRPO becomes even more critical in the more challenging low-data setting. The performance gap between GRPO and SFT increases from 4.72 mAP in the 10-shot scenario to 6.58 mAP in the 5-shot scenario. This widening margin highlights GRPO's ability to learn effectively in data-scarce environments where SFT struggles.

### 3.2.2 FEW-SHOT GENERALIZED REFERRING EXPRESSION SEGMENTATION

**Task Evaluation.** We conduct experiments on the EarthReason (Li et al., 2025b) dataset under few-shot setting. Following LISA (Lai et al., 2024), performance on the GRES task is measured

by the mask-based gIoU, defined by the average of all per-image IoUs. We use this metric because alternatives like cIoU are highly biased toward large-area objects and tend to fluctuate significantly. We report the final gIoU scores on the validation and test sets of the EarthReason dataset. To ensure a fair comparison with SFT-based approaches, we evaluate our method against SegEarth-R1 (Li et al., 2025b), trained on the same dataset. SegEarth-R1 serves as a strong SFT baseline, as it employs an auxiliary segmentation decoder to generate pixel-level masks through a differentiable mask loss.

**Results.**      In Table 4, we demonstrate the effective results of our proposed pipeline and task-specific reward for training reasoning models on GRES task. First, we found our GRPO-trained model, i.e., Geo-R1, demonstrates a significant improvement compared to the zero-shot baseline. It achieves a gIoU increase of up to 38.48% on the validation set (from 19.35% to 57.78%) and up to 26.11% on the test set (from 32.16% to 58.27%), showing the success of RL-based post-training paradigm. Then, we observe that the model exhibits remarkable performance with a very small number of samples. With just 240 samples (10-shot), our model demonstrates a comparable performance with PixelLM, which are trained on 900K instances with descriptions. *Using only 240 samples (10-shot), which is roughly 2% training data, Geo-R1 reaches nearly 83% of the performance of the SegEarth-R1 model that was trained on the entire training set.*

Table 4: Performance on the EarthReason for the GRES task. We report gIoU.

|  | Val | Test |
|---|---|---|
| **Full Amount Fine-tune** | | |
| LISA | 61.04 | 60.88 |
| PixelLM | 57.94 | 60.01 |
| PSALM | 66.61 | 68.30 |
| SegEarth-R1 | 68.60 | 70.75 |
| **Zero-shot Baseline** | | |
| Qwen2.5-VL w/ thinking | 19.35 | 32.16 |
| **1-shot Fine-tune (24 samples)** | | |
| SegEarth-R1 | 42.47 | 43.01 |
| Geo-R1 | 51.38 | 50.30 |
| **5-shot Fine-tune (120 samples)** | | |
| SegEarth-R1 | 45.37 | 45.46 |
| Geo-R1 | 54.73 | 56.01 |
| **10-shot Fine-tune (240 samples)** | | |
| SegEarth-R1 | 56.40 | 56.60 |
| Geo-R1 | 57.78 | 58.27 |

In a direct comparison, the GRPO pipeline consistently yields superior models to the SFT approach. Geo-R1 outperforms SegEarth-R1 in both the 10-shot and 5-shot settings. Crucially, *this performance gap becomes more significant as the amount of training data decreases*. This trend indicates that RL-based post-training paradigm is a more effective and sample-efficient method for adapting large VLM to this specialized, pixel-level task, especially in data-scarce scenarios.

## 4 DISCUSSION

In this section, we first compare the learning dynamics of SFT and GRPO, then examine cross-dataset generalization, the upper bound of few-shot learning, and the impact of model size. Unless otherwise specified, experiments are conducted on the VRSBench-FS dataset under the 10-shot setting.

### 4.1 LEARNING CURVE COMPARISON

We fine-tune Qwen2.5-VL-3B with both SFT and GRPO on the REC task using same batch size and evaluate checkpoints every 100 steps to sketch the learning curve. As shown in Figure 2, GRPO consistently outperforms SFT at every checkpoint, with an average gain of 9.74%. GRPO improves steadily, peaking around 400 steps, and remains strong until the end, whereas SFT oscillated within 37%–40%. GRPO achieves a clearly higher ceiling and stabilizes around 50%, indicating better training efficiency under few-shot setting.

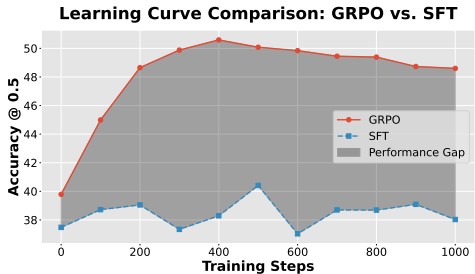

Figure 2: Learning curves of GRPO vs. SFT on REC.

## 4.2 CROSS DATASET GENERALIZATION

We further assess the cross-dataset generalization of the SFT and GRPO approaches on the GREC and GRES tasks. For the GREC task, we fine-tune models on the VRSBench dataset with limited supervision (1, 5, and 10-shot) and then evaluate model performance on the DIOR-RSVG target dataset, in a zero-shot manner. As shown in Table 5, GRPO consistently outper-

Table 5: Cross Dataset Evaluation.

| # shot | VRSBench → DIOR-RSVG | | EarthReason → RRSIS-D | |
|---|---|---|---|---|
| | SegEarth-R1 | Geo-R1 | SegEarth-R1 | Geo-R1 |
| 1-shot | 32.35 | 37.27 (+4.92) | 18.77 | 32.11 (+13.34) |
| 5-shot | 34.52 | 40.57 (+6.05) | 20.29 | 36.41 (+16.12) |
| 10-shot | 34.86 | 40.38 (+5.52) | 24.27 | 37.83 (+13.56) |

forms SFT across all settings, achieving a performance advantage of 4.92%, 6.05%, and 5.52% in the 1-shot, 5-shot, and 10-shot scenarios, respectively.

Similarly, for the GRES task, models were tuned on the EarthReason dataset (1, 5, and 10-shot) and tested on the RRSIS-D dataset. Here, the GRPO-based model (Geo-R1) demonstrates a remarkable improvement over the SFT-based model (SegEarth-R1) under few-shot setting, achieving a relative improvement up to 80%. These results highlight GRPO's incredible cross-dataset generalization, indicating superior transferability and robustness of Geo-R1.

## 4.3 UPPER BOUND OF FEW-SHOT LEARNING

As shown in Figure 3, GRPO clearly outperforms SFT in low-shot settings, although this performance gap narrows as supervision increases. To investigate this trend and determine the upper-bound capability of Geo-R1, we experimented with additional shot numbers (20, 50, 100, and 200). Concretely, the margin between GRPO and SFT approaches shrinks from 13.03% at 1-shot to 0.47% at 200-shot. This diminishing advantage suggests both approaches approach a common upper bound with more data. Empirically, they converge toward the full-data SFT result of 62.91%, indicating GRPO's strong sample efficiency at small shots but similar asymptotic performance as shot count grows.

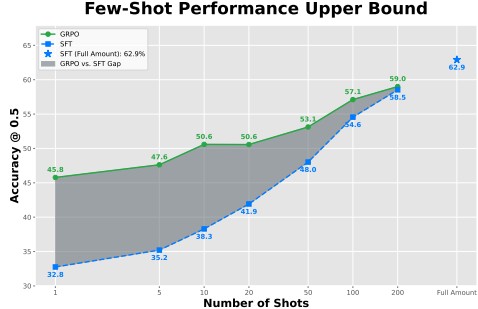

Figure 3: Few-shot Learning Upper-Bound.

## 4.4 FEW-SHOT LEARNING MEETS MODEL SIZE

We then examine how model size influences the performance under different post-training paradigms. As shown in Figure 4, both SFT and GRPO benefit from increased model scales. However, this trend exhibits clear diminishing marginal returns. For instance, SFT gains 4.31% when scaling from 3B to 7B but only 2.23% from 7B to 32B, with a similar slowdown observed for GRPO from 3B to 7B. This suggests that while larger models provide a stronger foundation, simply increasing number of parameters yields limited benefits for the few-shot task. This can be attributed to the limited fine-tuning data. With few examples, high-capacity models tend to overfit by simply memorizing the training samples rather

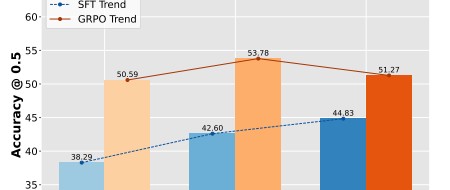

Figure 4: Few-shot Learning Meets Model Size.

than learning generalizable features. Notably, GRPO's performance decreased on the 32B model, likely due to two factors: overfitting on limited data and numerical instability from bf16 training.

## 5 RELATED WORK

**Reasoning LLMs and VLMs**. The OpenAI o1 (Jaech et al., 2024) showed that RL improves the reasoning capability of LLMs by learning from feedback on final outcomes. Recently, DeepSeek-

R1 (Guo et al., 2025) demonstrated that rule-based rewards can be used with the GRPO algorithm to teach LLMs advanced reasoning skills. Inspired by the success of RL in LLMs, researchers are now applying the R1 framework to VLMs. R1-OneVision (Yang et al., 2025) created a step-by-step multimodal reasoning datasets for SFT and RL. Concurrently, R1-V (Chen et al., 2025) applied the GRPO algorithm to object counting, achieving the remarkable result of a 3B model outperforming much larger 72B models. VisualThinker-R1-Zero (Zhou et al., 2025) applied it directly to base VLMs, observing "visual aha moments". Other studies refined the training process: Vision-R1 (Huang et al., 2025) first created a multimodal CoT dataset, serving as a cold-start before RL; LMM-R1 (Peng et al., 2025) used a two-phase strategy, starting with text-only reasoning before fine-tuning on multimodal data. Visual-RFT (Liu et al., 2025b), VLM-R1 (Shen et al., 2025), and Seg-Zero (Liu et al., 2025a) explored applying RL to image perception tasks.

**Few-shot Learning in Remote Sensing**. Few-shot learning (FSL) is crucial in RS, since it effectively addresses the challenge of limited labeled data. Attention-based contrastive learning have been shown to significantly improve classification accuracy in scene classification tasks (Xu et al., 2024; Zeng & Geng, 2022). Prototype-based networks (Li et al., 2021; Cheng et al., 2022) and multi-scale feature fusion strategies (Zhao et al., 2022) help models obtain diverse object characteristics, achieving state-of-the-art results on RS object detection benchmarks under few-shot settings. For segmentation, adaptive prototype clustering and mask-guided correlation learning enable precise pixel-level interpretation even with few annotated samples (Jiang et al., 2022; Jia et al., 2025; Li et al., 2024b; Shen et al., 2024). FSL enhances the efficiency and interpretability of RSI analysis, while also addressing key challenges in generalization and multimodal integration (Sun et al., 2021; Lee et al., 2024).

**REC and RES in Remote Sensing**. Referring expression comprehension in remote sensing—often termed remote sensing visual grounding (RSVG), which localizes a target in aerial imagery from a natural-language description. Early progress was established by the RSVG benchmark and the GeoVG model (Sun et al., 2022), and extended by DIOR-RSVG to broaden categories and scene scale (Zhan et al., 2023). In the MLLM era, GeoChat (Kuckreja et al., 2024) was the first MLLM to handle a wide range of RS vision-language tasks, including RSVG. Later, VRSBench (Li et al., 2024c) provided a high-quality dataset for RSVG task. RS-specific MLLMs such as EarthGPT (Zhang et al., 2024), RSGPT (Hu et al., 2025), SkySenseGPT (Luo et al., 2024), VHM (Pang et al., 2025), further unified different vision-language tasks, such as captioning, VG, VQA, and OVD, thus improving RS-specific alignment. For RES, Yuan et al. introduced the RES task for RS and released the RefSegRS dataset (Yuan et al., 2024). Liu et al. later introduced RRSIS-D, enabling pixel-level referring at scale (Liu et al., 2024d). Recent works such as GeoGround (Zhou et al., 2024b) and Skysense-O (Zhu et al., 2025) further unified the REC and RES tasks for RS images. Besides, works for OVD (Li et al., 2024e; Pan et al., 2025), and OVS (Li et al., 2025a;b) can be viewed as a special case of REC and RES (locate multiple objects with template-based description), which support grounding of novel categories.

## 6 CONCLUSION

In this work, we define a generic task, Referring Expression Understanding that aims to recognize objects (either detection/segmentation) from referring expressions. We then compare RL-based (GRPO) and SFT-based post-training paradigms on few-shot REC, OVD, and GRES tasks within the RS domain. Our results show that our GRPO-trained model, Geo-R1, consistently outperforms standard SFT-tuned models across these tasks. The performance gains are particularly large in low-shot regimes, and the model exhibits significantly stronger cross-dataset generalization.

While our study demonstrates the effectiveness of reinforcement learning for few-shot referring expression understanding, several avenues remain. Our evaluation is limited to high-resolution aerial imagery; extending Geo-R1 to multispectral (e.g., Sentinel-2) and SAR data would further test its robustness. Beyond the three REU tasks studied (REC, RES, OVD), future work could explore broader grounding tasks, e.g., OVS. Finally, scaling to larger shots, refining reward functions, and designing powerful RL training recipes remain promising directions.

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

## A  THE USE OF LARGE LANGUAGE MODELS

The LLMs were used in three ways: (i) to edit and polish grammar and phrasing; (ii) using "Deep-Research" to help retrieve and cluster related literature (with all citations verified by the authors). We reviewed, verified, and take full responsibility for the contents.

## B  REPRODUCIBILITY STATEMENT

We are committed to ensuring the reproducibility of our research. When we trained the SFT model, GRPO model and DAPO model, all the random seeds are fixed. Our implementation is built upon VLM-R1 and Easy-R1 codebase. All datasets used in our experiments, such as VRSBench, NWPU, EarthReason, RRSIS-D, and DIOR-RSVG, are publicly available. All models, training recipes will be open-sourced in `http://geo-r1.github.io` to make sure the results presented in our main paper are reproducible.

## C  PROMPT TEMPLATE

We largely follow the VLM-R1 prompt templates for REC, OVD, and extend the same interface to the GRES setting. We append the thinking template at the end for all task prompts.

---

**Prompt Template of REC**

*Please provide the bounding box coordinates of the region this sentence describes:* {`query`}.

---

**Prompt Template of OVD**

*Please carefully check the image and detect the following objects:* {`target list`}. *Output each detected target's bbox coordinates in JSON format. The format of the bbox coordinates is:*

```json
[
{
    "bbox_2d": [x1, y1, x2, y2],
    "label": "category name"
},
{
    "bbox_2d": [x1, y1, x2, y2],
    "label": "category name"
}
]
```

*If there are no such targets in the image, simply respond with None.*

---

**Prompt Template of GRES**

*Please carefully check the image and answer:* {`query`}. *Based on your answer, detect all relevant objects in the image. Output each detected target's bbox coordinates in JSON format. The format of the bbox coordinates is:*

```json
[
{

    "bbox_2d": [x1, y1, x2, y2],
    "keypoint1": [x3, y3],
    "keypoint2": [x4, y4]
},
{

    "bbox_2d": [x1, y1, x2, y2],
    "keypoint1": [x3, y3],
    "keypoint2": [x4, y4]
}
]
```

**Thinking Template**

{`problem`} *Output the thinking process in <think> </think> and final answer in <answer> </answer> tags.*

## D DEMO

We visualize the model's reasoning process for demonstration. Figure 5 shows some successful case on GRES task, while Figure 6 illustrates examples on GREC tasks. In general, Geo-R1 is able to provide a reasonable thinking process that leads to the accurate answer.

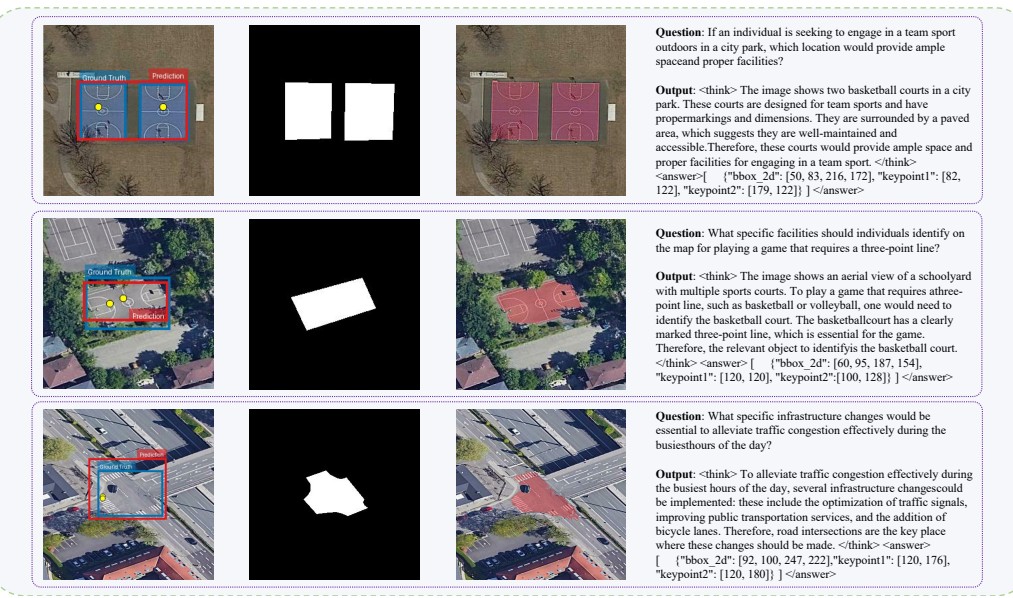

Figure 5: Geo-R1 inference samples (success case for GRES).

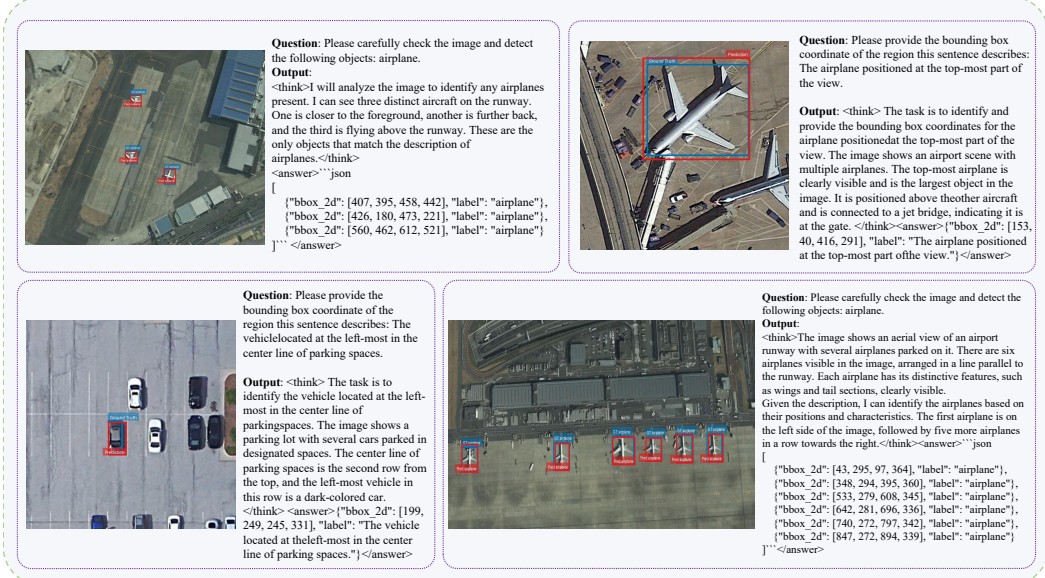

Figure 6: Geo-R1 inference samples (success case for GREC).

In general, Geo-R1 is able to provide a reasonable thinking process that leads to an accurate answer. However, we also identify key failure modes that highlight a disconnect between an often-plausible reasoning process and the final spatial output.

As shown in the top-left of Figure 7, the model can correctly reason in its reasoning process that the park area is the most suitable region for wildlife preservation, yet its final answer incorrectly grounds this concept to the nearby sports facility. In the bottom-right example, the model's reasoning correctly identifies the presence of multiple ships, but the final answer detects only the cluster on the left while omitting the group on the right. Furthermore, we also observe failures in reasoning process, the bottom-left example shows a failure within the reasoning process itself. Although the model accurately understands the task in its reasoning block (to identify the 'large vehicle'), it subsequently fails to apply this critical size attribute, incorrectly selecting a much smaller, adjacent vehicle for its final answer. These cases suggest that a key area for future improvement is strengthening the alignment between the model's high-level semantic understanding and the generation of accurate, complete, and precise spatial coordinates.

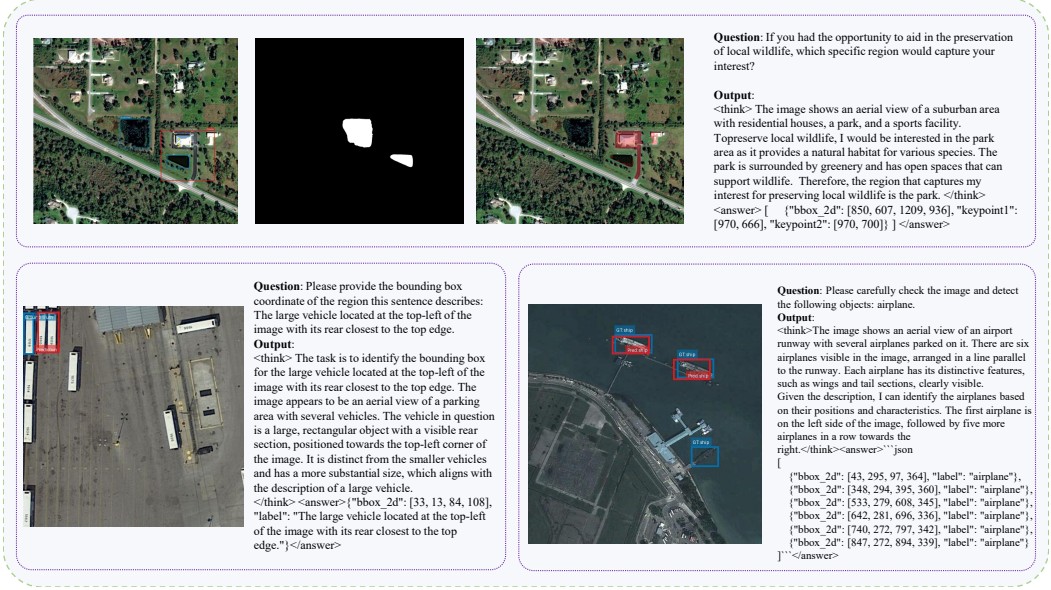

Figure 7: Geo-R1 inference samples (failure case).

# E  ADDITIONAL EXPERIMENTS

## E.1  ABLATION STUDY ON FORMAT REWARD

Table 6 shows that without the format reward, Geo-R1 exhibits slightly degraded performance on both REC and GRES tasks. Empirical results also show that Geo-R1 w/o format reward requires much longer training time to converge (about 1.6× longer for REC, 1.7x longer for OVD and 1.3x longer for GRES) compared to Geo-R1 w/ format reward.

Table 6: Ablation on Format Reward.

|  | REC (Acc@0.5) | GRES-val (gIoU) | GRES-test (gIoU) | OVD (mAP) |
|---|---|---|---|---|
| Geo-R1 w/ format reward | 50.59 | 57.78 | 58.27 | 34.92 |
| Geo-R1 w/o format reward | 48.23 | 56.61 | 57.24 | 34.64 |

## E.2  ABLATION STUDY ON PENALTY OF OVERLONG OVD PREDICTION RESULT

Table 7 shows that without the length penalty reward, Geo-R1 exhibits significantly degraded performance on the OVD task.

Table 7: Ablation on Length Penalty of OVD task.

|  | OVD (mAP) |
|---|---|
| Geo-R1 w/ length penalty | 34.92 |
| Geo-R1 w/o length penalty | 27.52 |

## E.3  VARIANCE AND ROBUSTNESS REPORT

The few-shot training samples were randomly selected. To further assess robustness with respect to shot selection, we re-sampled the training data using different random seeds and re-ran the experiments for the REC and GRES tasks. For OVD, we re-sampled with 3 different random seeds due to time constraint and report 5-shot result (10-shot setting for OVD is hard to re-sample different samples). As shown in the Table 8, our model exhibits stable performance across 10 different seeds, with standard deviations below 1%.

Table 8: Robustness with Different Few-shot Samples.

| Task | Metrics | Reported (Seed 42) | Average ± std | Seed 43 | Seed 44 | Seed 45 | Seed 46 | Seed 47 | Seed 48 | Seed 49 | Seed 50 | Seed 51 | Seed 52 |
|---|---|---|---|---|---|---|---|---|---|---|---|---|---|
| REC | Acc@0.5 | 50.59 | 49.46 ± 0.95 | 49.90 | 49.76 | 50.34 | 49.77 | 50.28 | 47.43 | 49.20 | 49.61 | 50.33 | 47.99 |
| GRES-Val | gIoU | 57.78 | 59.19 ± 0.31 | 59.08 | 58.78 | 59.41 | 59.58 | 59.57 | 58.80 | 59.33 | 59.52 | 58.89 | 58.94 |
| GRES-Test | gIoU | 58.27 | 58.05 ± 0.63 | 59.00 | 58.20 | 58.18 | 57.27 | 58.41 | 57.10 | 58.47 | 58.59 | 58.17 | 57.11 |
| OVD | mAP | 33.20 | 34.29 ± 0.45 | 34.36 | 33.71 | 34.79 | | | | | | | |

## E.4  ADDITIONAL RESULT FOR REC TASK (OPT-RSVG DATASET)

To validate the effectiveness of Geo-R1, we further test the Geo-R1 on OPT-RSVG (Li et al., 2024a) dataset under a zero-shot setting. SFT baseline is compared, and we report the Acc@0.5. Result can be seen in Table 9. From this Table, our Geo-R1 model shows significantly better performance than the SFT counterpart.

Table 9: REC Result on OPT-RSVG Dataset.

|        | Val   | Test  |
|--------|-------|-------|
| SFT    | 30.24 | 31.06 |
| Geo-R1 | 33.76 | 34.29 |

### E.5    THINKING V.S. NOT THINKING

We provide RFT results with and without thinking in Table 10. Overall, RL training with thinking provides slightly better performance and much better interpretability. Moreover, RFT consistently outperforms SFT regardless of whether thinking is applied.

Table 10: Comparison on models with and without Thinking

|                     | REC (Acc@0.5) | OVD (mAP) |
|---------------------|---------------|-----------|
| SFT                 | 38.29         | 30.20     |
| Geo-R1 w/ thinking  | 50.59         | 34.92     |
| Geo-R1 w/o thinking | 48.10         | 32.79     |

### E.6    PARAMETER EFFICIENT FINETUNING FOR GEO-R1

we train our Geo-R1 model using LoRA (rank = 64, alpha = 128), and the results are shown in the Table 11 below. Results show that combining LoRA with Geo-R1 slightly hurts the performance and does not save much training time compared with full fine-tuning (9 hours vs. 10 hours). This is because the rollout operation dominates the GRPO training process, while updating the LLM accounts for only a small portion of the total time.

Table 11: Parameter Efficient Finetuning for Geo-R1.

|                     | REC (Acc@0.5) | GRES-val (gIoU) | GRES-test (gIoU) |
|---------------------|---------------|-----------------|------------------|
| Geo-R1 LoRA         | 47.62         | 51.46           | 53.89            |
| Geo-R1 full-finetune| 50.59         | 57.78           | 58.27            |

### E.7    COMPARISON WITH OTHER APPROACHES.

We add a comparison, Grounding-DINO (Liu et al., 2024c), for REC task. Specifically, the Grounding-Dino-T is fine-tuned with 10-shot and 5-shot REC data for 30 epoch, using Open-Grounding-Dino framework [4]. The result can be seen in Table 12.

Table 12: Comparison with Other Approaches.

|                | 10-shot | 5-shot | 1-shot |
|----------------|---------|--------|--------|
| Grounding-DINO | 37.65   | 26.52  | 22.51  |
| Qwen2.5-VL-SFT | 38.29   | 35.21  | 32.75  |
| Geo-R1         | 50.59   | 47.62  | 45.78  |

---

[4]https://github.com/longzw1997/Open-GroundingDino

## E.8 Assessment on Generated Reasoning Chains

To further validate reasoning quality, we conducted an additional analysis in which we prompted Qwen3-VL-235B-A22B-Thinking [5] to automatically evaluate a randomly sampled subset of reasoning chains whose corresponding predictions have IoU $> 0.5$ on the VRSBench test set (1500 samples). We carefully designed prompts to assess both the correctness and usefulness of the reasoning toward the final prediction, assigning score on a 1–10 scale. As summarized in the Figure 8, the average reasoning quality score is 8.03 with std of 1.47, indicating that the majority of generated reasoning chains are reasonable, informative, and supportive of the final predictions.

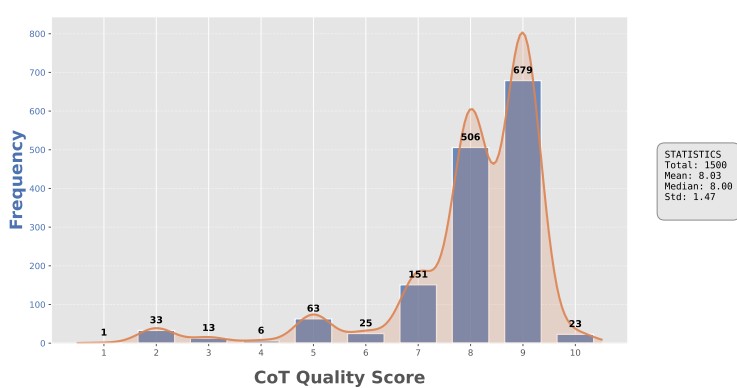

Figure 8: Statistics of Quality Assessment on Generated Reasoning Chains.

---

**CoT Quality Evaluation Prompt**

**You are a strict evaluator** for a vision–language model that performs **referring expression comprehension (visual grounding)** on remote sensing images.

For each sample, you will be given:

- `image`: a remote sensing / aerial image.
- `question`: a natural–language referring expression describing a target region.
- `ground_truth`: the correct bounding box of the referred region: `[x_min, y_min, x_max, y_max]`.
- `model_output`: the model's full output, which contains:
  - a reasoning **Chain-of-Thought** between `<think> ... </think>`,
  - a final JSON-style prediction between `<answer> ... </answer>` with a predicted bbox `bbox_2d`.
- `extracted_answer`: the predicted bbox `[x_min, y_min, x_max, y_max]` parsed from the final answer.
- `correct`: whether the final prediction is counted as correct (1 or 0).

Your job: **only evaluate the reasoning text inside `<think>...</think>`** and output **a single overall quality score from 1 to 10** (higher is better).
You should internally consider two aspects:

1. **Correctness** of the reasoning.
2. **Usefulness** of the reasoning.

But your final output must be just **one combined score** that reflects both.
Do **not** change the model's prediction. Do **not** generate a new bounding box.

---

[5] https://github.com/QwenLM/Qwen3-VL

**1. Criterion to Consider When Scoring**

**1.1. Correctness**

Correctness measures how **factually accurate and logically sound** the reasoning is, relative to:

- the image content,
- the question / referring expression,
- the ground-truth bounding box and the predicted bounding box.

Key points:

1. **Visual faithfulness**

   - The reasoning should accurately describe what is visible (object categories, spatial relations, comparative relations, etc.).
   - Penalize:
     - referring to objects that do not exist in the image,
     - clearly wrong locations (e.g., calling a top–left object "bottom–right"),
     - confusing object types (e.g., calling a tennis court a baseball field).

2. **Consistency with ground-truth and prediction**

   - Use `ground_truth` and `extracted_answer` to understand which region is correct and how close the prediction is.
   - If the predicted bbox is near the ground truth and the reasoning clearly supports this localization, correctness should be **high**, as long as there are no serious hallucinations.
   - If the predicted bbox is far from ground truth and the reasoning is based on the wrong object/region, correctness should be **low**, even if the text is fluent.

3. **Logical consistency**

   - The reasoning should be internally coherent:
     - no self-contradiction (e.g., first says "top–left" then "bottom–right" for the same object),
     - no obvious contradiction with the predicted bbox (e.g., reasoning says the object is "top–left" but the predicted box is in bottom–right).

**Rule:** If correctness is extremely low (e.g., the reasoning clearly focuses on a completely wrong object or is mostly hallucinated), the final overall score must also be low (around 1–3), no matter how nicely written it is.

**1.2. Usefulness**

Usefulness measures how **helpful and informative** the reasoning is for explaining **why** the model chose the final bounding box.

Key points:

1. **Task-specific grounding**

   - A useful CoT clearly connects the question to the image:
     - identifies candidate objects that match the category (ships, vehicles, fields, buildings, etc.),
     - compares positions (left-most, bottom-most, near center, near boundary, etc.),
     - uses context (e.g., "among the ships", "in the bottom-left corner", "between two runways", "inside the stadium", "on the water, not on land").
   - Penalize:
     - CoT that just paraphrases the question without adding visual evidence,
     - very generic lines ("I look at the image and find the object described") with no actual reasoning.

2. **Conciseness and relevance**

   - Prefer reasoning that is focused on the grounding task.

- Penalize:
  - very long but repetitive reasoning that adds no extra information,
  - completely off-topic storytelling.

---

**2. How to Combine into a Single Score (1–10)**

You must convert your internal judgment about correctness and usefulness into a **single integer score from 1 to 10**.

Use the following guidelines:

- **9–10 (Excellent overall quality)**: reasoning is highly correct (no major factual or spatial mistakes, no hallucinations, strongly aligned with the correct region) and highly useful (explicitly ties language to visual evidence, considers candidates, compares positions, and explains why the final region is chosen). This is the kind of CoT you would want as a **gold standard teaching signal**.

- **7–8 (Good overall quality)**: reasoning is mostly correct, with at most minor inaccuracies or slight vagueness. It is useful but may miss some steps or be less detailed (e.g., fewer explicit candidate comparisons). Still clearly grounded and helpful.

- **4–6 (Mixed / moderate quality)**: reasoning is partially correct (right general area or object type) but has notable gaps, imprecision, or some incorrect statements. Usefulness is limited: some connection between question and image exists, but the explanation is shallow, generic, or incomplete. This range also includes cases where the CoT is well structured but correctness is only moderate.

- **1–3 (Poor overall quality)**: reasoning is mostly incorrect or hallucinated, focusing on the wrong object or contradicting obvious visual evidence, or is so vague/generic that it provides almost no real grounding. Even if the final predicted bbox happens to be correct, if the CoT is wrong or useless, the score must remain low.

---

**3. Evaluation Procedure**

For each sample:

1. **Understand the target:** read the question to understand what region is being referred to (e.g., "left-most ship", "top-center soccer field", "vehicle at bottom-right", "bridge in the middle").

2. **Inspect ground-truth and prediction:** use `ground_truth` and `extracted_answer` to understand which region is correct and which region was chosen by the model. This helps you judge if the reasoning aligns with the proper region.

3. **Read the CoT inside `<think>...</think>`:** ignore anything outside `<think>...</think>` for scoring. Evaluate how correct and how useful this reasoning is as described above.

4. **Assign a single overall score (1–10):** combine correctness and usefulness as described in Section 2. Also provide a brief explanation of why you chose this score.

---

**4. Output Format**

Return your evaluation as a **Python-style dictionary literal**, with no extra text:

```
{
  "score":  <integer 1-10>,
  "explanation":  "<1-3 short sentences explaining your
judgment of correctness and usefulness together>"
}
```

