Building upon the foundation of reinforcement learning algorithms like Proximal Policy Optimization (PPO) (Schulman et al., 2017), Group Relative Policy Optimization (GRPO) (Shao et al., 2024) has become a more effective and memory-efficient evolution for training advanced reasoning models by eliminating the value model and changing the reward model to rule-based reward. The advantage is estimated in a group-relative manner, simplifying the training process.

The GRPO algorithm begins by sampling N candidate outputs $\{o_1, \ldots, o_N\}$ from the current policy $\pi_\theta$ for a given query prompt $q$. Each response $o_i$ is then evaluated by a reward function $R(q, o_i)$ to obtain a raw reward score $r_i$. To measure the relative quality of each response within the sampled group, GRPO standardizes the raw rewards to obtain the advantage value, as shown in Eq. 1. The advantage value $\hat{A}_i$ denotes the normalized advantage of the response $o_i$ relative to other samples within the group.

$$\hat{A}_i = \frac{r_i - \text{mean}\{r_1, r_2, \ldots, r_N\}}{\text{std}\{r_1, r_2, \ldots, r_N\}} \tag{1}$$

The policy $\pi_\theta$ is updated with a training objective (Eq. 2), designed to encourage the generation of responses with higher advantages.

$$\mathcal{J}_{\text{GRPO}}(\theta) = \mathbb{E}_{\{o_i\}_{i=1}^N \sim \pi_{\theta_{\text{old}}}(\cdot|q)} \left[ \frac{1}{N} \sum_{i=1}^N \left( \min\left(c_1 \cdot \hat{A}_i, \ c_2 \cdot \hat{A}_i\right) - \beta D_{\text{KL}}(\pi_\theta || \pi_{\text{ref}}) \right) \right], \tag{2}$$

where

$$c_1 = \frac{\pi_\theta(o_i \mid q)}{\pi_{\theta_{\text{old}}}(o_i \mid q)}, c_2 = \text{clip}\left(\frac{\pi_\theta(o_i \mid q)}{\pi_{\theta_{\text{old}}}(o_i \mid q)}, 1 - \varepsilon, 1 + \varepsilon\right). \tag{3}$$

Here, $D_{\text{KL}}(\pi_\theta || \pi_{\text{ref}})$ denotes the KL divergence between the current policy $\pi_\theta$ and the reference policy $\pi_{ref}$, which serves as a regularization term to prevent large deviations. The clipping mechanism in $c_2$ stabilizes training by constraining the policy update ratio.

For LLMs on tasks with definitive answers, like mathematical reasoning, this reward can be calculated using a rule-based verifiable reward function. By applying a reward to the thinking process and using the GRPO algorithm, DeepSeek-R1 (Guo et al., 2025) can provide both final answers and the logical reasoning process that leads to them. This approach can be extended to VLMs by converting visual metrics into tailored reward signals. Recently, there are some works that adapt GRPO for vision-language tasks by engineering task-specific rewards, such as VLM-R1 (Shen et al., 2025), Visual-RFT (Liu et al., 2025b), Seg-Zero (Liu et al., 2025a), etc.

### 2.2 FEW-SHOT REFERRING EXPRESSION UNDERSTANDING TASK

Referring Expression Understanding (REU) serves as a unifying framework for Generalized Referring Expression Comprehension (GREC) and Generalized Referring Expression Segmentation (GRES). Given an image $I$ and a textual query $q$, a vision–language model (VLM) $\mathcal{F}$ predicts one or more target objects, as formulated in Eq. 4:

$$\{O_1, \ldots, O_N\} = \mathcal{F}(I, q), \tag{4}$$

where each $O_i$ denotes a predicted object parsed from VLM text outputs, and $N$ denotes the number of parsed objects. In the case of GREC, $O_i$ is represented as a bounding box, while for GRES, $O_i$ is represented as an instance mask.

Tasks such as REC, RES, Generalized REC (GREC) (He et al., 2023), Generalized RES (GRES) (Liu et al., 2023), Visual Grounding (VG) (Plummer et al., 2015), Open-Vocabulary Detection

(OVD), and Open-Vocabulary Segmentation (OVS) (Wu et al., 2024) can all be seen as specialized forms of REU, each with a different emphasis.

In this work, we focus on three representative REU tasks: (i) REC, which targets single-object detection from complex reasoning queries; (ii) OVD, which addresses multi-object detection from template-based queries; and (iii) GRES, which requires multi-object segmentation from complex reasoning queries. All tasks are studied under few-shot settings. In our formulation, the number of shots refers to the number of annotated bounding boxes or masks. Specifically, in the FS-GREC setup, including FS-REC and FS-OVD, one "shot" is defined as an image–query–box triplet, while in FS-GRES, one "shot" corresponds to an image–query–mask triplet. Importantly, a ground-truth mask may include multiple valid instances for a single query (Li et al., 2025b). Among these tasks, FS-GRES is the most challenging, as it constitutes a pixel-level extension of FS-GREC and

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

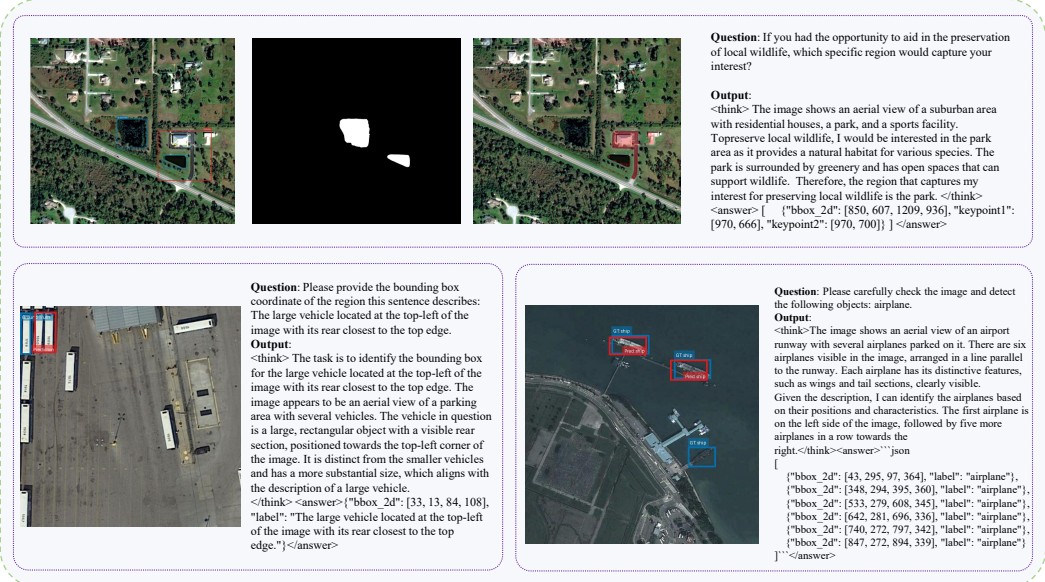

Figure 7: Geo-R1 inference samples (failure case).