# OpenReview forum: "Geo-R1: Improving Few-Shot Geospatial Referring Expression Understanding with Reinforcement Fine-Tuning"
_ICLR.cc/2026/Conference — ICLR 2026 Conference Withdrawn Submission_

### Official Review · Reviewer_DU3H · 2025-10-26

**Soundness:** 3
**Presentation:** 3
**Contribution:** 2
**Rating:** 4
**Confidence:** 5

**Summary:**

This paper addresses the challenges of Referring Expression Understanding (REU) in remote sensing images under few-shot settings by proposing a novel method named Geo-R1. The approach leverages GRPO-based reinforcement learning that performs reasoning before action. Experimental results on three distinct REU datasets demonstrate the effectiveness of the proposed method and highlight its strong cross-dataset generalization capability.

**Strengths:**

1. Inspired by DeepSeek-R1-Zero, the paper adapts the “reason-then-act” GRPO reinforcement learning strategy to the remote sensing REU domain. Remarkably, it achieves strong performance even with extremely limited dataset (e.g., 1-shot).
2. The paper provides thorough experimental evaluations, including learning curves, cross-dataset generalization, the impact of training data scale, and model size.

**Weaknesses:**

1. The work is primarily engineering-oriented, focusing on the application of general-purpose methods to remote sensing tasks, with limited theoretical or methodological innovation.
2. The quality of the generated reasoning chains is not assessed. The DeepSeek-R1-Zero–style training paradigm imposes no explicit constraints on the reasoning process itself. Notably, Figure 7 includes examples where reasoning appears correct but leads to misaligned localization, or where the reasoning itself is flawed—highlighting the need for dedicated reasoning evaluation.
3. Since GRPO requires generating multiple responses per training instance, its computational cost is substantially higher than standard supervised fine-tuning (SFT). Therefore, comparing primarily against SFT baselines may not be equitable. A more appropriate comparison would involve other few-shot learning approaches that operate under similar resource constraints.

**Questions:**

1. Were the few-shot training examples selected randomly, or were they curated to possess certain characteristics (e.g., diversity, difficulty, or representativeness)?
2. How is the reward computed for outputs involving multiple bounding boxes or keypoints? For instance, in GRES, if the model outputs {..., "keypoint1": [82, 122], "keypoint2": [179, 122]} while the ground truth is {..., "keypoint1": [179, 122], "keypoint2": [82, 122]}, or if the prediction is a single large bounding box encompassing two smaller ground-truth boxes—how is alignment and reward determined in such cases?
3. To the best of my knowledge, GRPO-based methods typically require training data on the order of thousands of examples. Is this work the first to successfully apply GRPO in a true few-shot data(e.g., 1-shot or 5-shot)?
4. It is recommended that the authors complement the performance on OPT-RSVG[1] to further validate the generalization of the proposed method

[1] Language-guided progressive attention for visual grounding in remote sensing images[J]. IEEE Transactions on Geoscience and Remote Sensing, 2024, 62: 1-13.

---

> ### Author Response · Authors · 2025-11-21
> **To Assigned Reviewer PFsJ (Reviewer 4)**
>
> - **C4.1**: Concerns about method novelty.
>
>      - **R4.1**: We believe there is a misunderstanding regarding the core contributions of our work. Our paper does not simply apply GRPO to remote sensing datasets. Instead, we introduce a **new task setting—few-shot geospatial referring expression understanding**—and develop **task-specific reward designs** tailored to this setting. These contributions were explicitly recognized by other reviewers. For example, Reviewer PFsJ noted that our work “introduces a reasoning-centric reinforcement fine-tuning (RFT) framework for few-shot geospatial referring expression understanding” and “develops task-aligned reward functions (IoU, mAP, MaskGIoU) tailored to spatial grounding, including an innovative non-differentiable ‘BBox + SAM’ pipeline for segmentation”. Reviewer T8hh further emphasized that “the application of the RL paradigm to the few-shot remote sensing scene is a valuable and relevant direction for the field” and that the paper “successfully extends from simpler REC and RES tasks to the more complex GREC and GRES tasks”.
>       - In addition, we do not claim to introduce a new dataset; rather, we define and study a new task—few-shot geospatial referring expression understanding—grounded in remote sensing imagery. This task involves unique spatial reasoning challenges and requires specialized reward design and training strategies, which our work proposes and evaluates.
>
> - **C4.2**: Questions about the quality of the generated reasoning chains.
>
>      - **R4.2**: Thank you for raising this insightful point. We acknowledge that our work has not systematically evaluated the quality of the generated reasoning chains. This is primarily due to the lack of high-quality CoT annotations and the absence of standardized evaluation protocols for reasoning quality in the remote sensing domain. In our preliminary analysis, we randomly sampled model outputs and conducted manual inspection, finding that most reasoning chains were generally coherent and aligned with the visual content, despite occasional mismatches between reasoning and localization (as illustrated in Figure 7).
>      - To further validate reasoning quality, we conducted an additional analysis in which we prompted [Qwen3-VL-235B-A22B-Thinking](https://github.com/QwenLM/Qwen3-VL) to automatically evaluate a randomly sampled subset of reasoning chains whose corresponding predictions have IoU > 0.5 on the VRSBench test set (1500 samples). We carefully designed prompts to assess both the correctness and usefulness of the reasoning toward the final prediction, assigning a score on a 1–10 scale. The mean reasoning quality score is 8.03 with std of 1.47 (**see section E.6 of the updated pdf** for more details), indicating that the majority of generated reasoning chains are reasonable, informative, and supportive of the final predictions.

---

> ### Author Response · Authors · 2025-11-21
> **Continue**
>
> - **C4.3**: Question about computational cost and comparison with other few-shot learning approaches.
>
>   - **R4.3**: First, we acknowledge that our RL-based training requires more computational cost compared to SFT. Geo-R1 is designed to address the challenges of high annotation costs and data scarcity (few-shot) in the remote sensing domain. As we prove in our paper, GRPO yields significant improvements in data efficiency compared to SFT. The increased compute cost is thus a worthwhile trade-off for the substantial reduction in data and annotation cost. Despite this increase, the total training time remains manageable (typically less than one day). Importantly, the inference cost of a GRPO-trained model remains comparable to that of SFT.
>
>    - Second, we note that the fundamental limitation of SFT-based methods for the referring expression understanding task stems from the next-token-prediction (NTP) nature of LLMs, which does not inherently support coordinate regression. Most of the existing few-shot learning approaches, such as LoRA, adapter tuning, and visual prompt tuning—despite their parameter efficiency—remain within the SFT regime and inherit the same training-objective drawback. In contrast, our RL-based training framework enables geometry-aware optimization through task-specific rewards (e.g., box AP or mask GIoU).
>
>    - We add a comparison, Grounding-DINO. Specifically, the Grounding-Dino-T is fine-tuned with 10-shot, 5-shot and 1-shot REC data for 30 epoch, using [Open-Grounding-Dino](https://github.com/longzw1997/Open-GroundingDino) framework.
>
>        || 10-shot | 5-shot |1-shot|
>        |:-:|:-:|:-:|:-:|
>        |Grounding-DINO| 37.65 | 26.52|22.51|
>        |Qwen2.5-VL-SFT | 38.29 |35.21|32.75|
>        |Geo-R1 |50.59  |47.62|45.78|
>
>
> - **C4.4**: Question about few-shot training examples selection and robustness?
>
>      - **R4.4**: The few-shot training samples were randomly selected. To further assess robustness with respect to shot selection, we re-sampled the training data using 10 different random seeds and re-ran the experiments for the REC and GRES tasks. For OVD, we re-sampled with 3 different random seeds due to time constraint and report 5-shot result (10-shot setting for OVD is hard to re-sample different samples). As shown in the table below, our model demonstrates stable performance across seeds, with standard deviations below 1%.
>
>     | Task |Metrics| Result from the Paper (Seed 42)| Average $\pm$ std|Seed 43 | Seed 44 | Seed 45 | Seed 46 | Seed 47 | Seed 48 | Seed 49 | Seed 50 | Seed 51 | Seed 52 |
>     | :-: | :-: | :-: | :-: |:-: | :-: | :-: | :-: | :-: | :-: | :-: | :-: | :-: | :-: |
>     |REC|Acc@0.5|50.59|49.46 $\pm$ 0.95|49.90|49.76|50.34|49.77|50.28|47.43|	49.20|	49.61|	50.33|	47.99|
>     |GRES|GIoU (Val)|57.78|59.19 $\pm$ 0.31|59.08	|58.78	|59.41|	59.58|	59.57	|58.80	|59.33|59.52|58.89|58.94|
>     |GRES|GIoU (Test)|58.27|58.05 $\pm$ 0.63|59.00|	58.20|	58.18|	57.27	|58.41|57.10|58.47|58.59|58.17|	57.11|
>     |OVD|mAP|33.20|33.68 $\pm$ 0.97|34.36|33.71|34.79|33.50|32.10|34.20|34.74|32.09|32.87|34.48|

---

> ### Author Response · Authors · 2025-11-21
> **Continue**
>
> - **C4.5**: Questions about reward calculation, especially for multiple instance predictions.
>
>     - **R4.5** First, we emphasize that our model is capable of predicting multiple objects (boxes or masks), and the reward is formulated in a many-to-many manner. For GRES, the reward is computed using the mask gIoU between the predicted mask and the ground-truth mask. To obtain the predicted mask, we feed all predicted “bounding box + keypoint” groups into SAM2, generate the corresponding instance masks, and then take their union to form a single predicted segmentation. This aggregated mask is then compared with the ground-truth mask via gIoU, making the reward invariant to permutations of keypoints or object ordering. For this task, even if the prediction is a single large bounding box encompassing two smaller ground-truth boxes, the keypoint predictions can still guarantee that the SAM2 model outputs multiple correct instance masks.
>           - For OVD, we follow standard practice and compute the mAP between the set of predicted boxes and the set of ground-truth boxes, which naturally handles many-to-many matching through IoU-based assignment. For this task, if the prediction is a single large bounding box encompassing two smaller ground-truth boxes, the reward would be low according to mAP calculation.
>
> - **C4.6**: Is this work the first to successfully apply GRPO in a true few-shot data?
>
>     - **R4.6** Yes, our work is the first to explore RFT-based few-shot adaptation of VLMs in the domain of remote sensing. Moreover, we are probably the first to explore RFT-based VLM for GRES task in the broader computer vision domain.
>
> - **C4.7**: Results on  OPT-RSVG.
>
>     - **R4.7** Thanks for the reviewer's suggestion. We conducted experiments on the OPT-RSVG dataset, and the results are given below. SFT baseline is compared, and we report the Acc@0.5.
>
>       ||Val | Test|
>       |:-:|:-:|:-:|
>       |SFT | 30.24 |31.06|
>       |Geo-R1|33.76|34.29 |

---

> > ### Comment · Reviewer_DU3H · 2025-11-28
> > **comment**
> >
> > After carefully reading the authors’ response, I agree that this paper presents interesting and practically valuable results for the geospatial AI and remote sensing communities. The empirical findings, datasets, and application scenarios are meaningful, and I can imagine this work being well received at a domain-specific venue.
> >
> > However, I also share Reviewer srr5’s concerns about the level of novelty from a machine learning perspective. While the application to remote sensing is certainly relevant and useful, the work does not appear to introduce new theoretical insights or clear algorithmic innovations. In addition, I would suggest that the authors include the zero-shot results provided during the rebuttal period in the main paper, as they help to strengthen the empirical evaluation.
> >
> > In summary, although I believe this work is valuable for the remote sensing community, I do not think it fully meets the innovation threshold typically expected for ICLR. Therefore, I cannot recommend acceptance, but I encourage the authors to consider submitting to a more specialized venue, where their contributions are likely to be highly appreciated.

---

> > > ### Author Response · Authors · 2025-11-28
> > > **Reply**
> > >
> > > Thank you for the suggestion. We will consider submit it to a domain-specific venue.

---

### Official Review · Reviewer_T8hh · 2025-10-28

**Soundness:** 3
**Presentation:** 2
**Contribution:** 2
**Rating:** 4
**Confidence:** 4

**Summary:**

The paper introduces Geo-R1, a novel Reinforcement Learning (RL) post-training methodology aimed at enhancing Geospatial Referring Expression Understanding (REU) in few-shot settings. The study focuses on three representative REU tasks: Referring Expression Comprehension (REC), Open-Vocabulary Detection (OVD), and Generalized Referring Expression Segmentation (GRES). By employing task-specific reward functions, Geo-R1 is designed to generate explicit intermediate reasoning hypotheses, thereby iteratively optimizing spatial localization and consequently improving the model's generalization capabilities under data-scarce conditions. Empirical results indicate that models fine-tuned with Geo-R1 surpass standard Supervised Fine-Tuning (SFT) models across various benchmarks, particularly demonstrating stronger cross-dataset generalization with limited sample sizes.

**Strengths:**

1. The application of the RL paradigm to the few-shot remote sensing scene is a valuable and relevant direction for the field.

2. The work successfully extends from simpler REC and RES tasks to the more complex GREC and GRES tasks. Furthermore, compared to methods like Seg-Zero, this approach achieves end-to-end segmentation optimization. While existing works (e.g., LENs) have explored similar end-to-end structures, this is still a strong contemporary contribution.

**Weaknesses:**

1. The paper suffers from significant redundancy and conceptual ambiguity. The frequent and repeated mentions of "few-shot" are excessive and feel redundant, especially since the improved performance with fewer samples using an RL paradigm is a generally acknowledged finding. More critically, the definition and scope of the core task, REU, are inconsistent: on Line 161, it is stated that REC, RES, GREC, GRES, VG, OVD, and OVS are all REU tasks, yet on Line 152, REU is defined as a combination of GREC and GRES. This lack of clarity and conflicting definitions makes the paper difficult to parse. Similarly, the detailed definitions in Lines 165–174 are largely redundant. Moreover, dedicating nearly an entire page of Section 2.1 to explaining the GRPO algorithm, which is a well-established and widely accepted method, significantly detracts from the paper's novel contributions.

2. I have serious reservations regarding the experimental results presented in Table 3. Under the low-data (few-shot) condition, the perplexing finding is that only the GTF metric shows an improvement, while performance on all other metrics inexplicably declines. This counter-intuitive result requires extensive justification. Additionally, I need clarification on the training details: What is the data repetition (data repeat/epoch) count for the available training samples in both the RL and SFT procedures? If the iteration count is $1$ in both cases, the models likely have not converged, which would fundamentally invalidate the experimental comparison.

3. To genuinely validate the real effectiveness of the proposed RL paradigm, a full-scale experimental comparison is mandatory. The current results are insufficient; the authors must provide a comparison against full-SFT models trained with full datasets to demonstrate the ultimate potential and effectiveness of Geo-R1.

**Questions:**

Pertaining to Figure 1, the definition provided, "1-shot == 26 samples," appears to be a highly unconventional use of the 'shot' terminology. Is this a specific, ad-hoc definition adopted for the purpose of this paper? Could the authors please provide a detailed and rigorous justification for selecting precisely 26 samples to represent a single "shot"?

---

> ### Author Response · Authors · 2025-11-21
>
> **C3.1 Redundancy and conceptual ambiguity.**
>
> **R3.1**: Thank you for the constructive suggestions regarding writing and presentation. We apologize for any confusion and have revised the paper accordingly to reduce redundancy and improve conceptual clarity.
> - First, in the previous version, we explicitly used the full terminology to emphasize that all experiments were conducted in a few-shot setting. In the revised manuscript, we have removed redundant “FS–” prefixes to streamline the presentation.
> - Second, the definition and scope of REU in our work are consistent. We define REU as a unified framework for object grounding (detection or segmentation) from referring expressions, which encompasses generalized forms of REC and RES. Under this view, REC, VG, and OVD are special cases of generalized form of REC with different focuses, while RES and OVS fall under generalized form of RES. Lines 165–174 specify how REU is instantiated under the few-shot setting. To eliminate ambiguity, we have rewritten Section 2.2 to provide a clearer and more precise formulation.
> - Finally, while we recognize the reviewer’s concern regarding the length of Section 2.1, we believe that a concise introduction to GRPO is necessary for readers in the remote sensing community who may be unfamiliar with this algorithm. Nonetheless, following the reviewer’s feedback, we have rewritten and compressed this section to avoid distracting from the core contributions of the paper.
>
> **C3.2 Questions about experimental results presented in Table 3.**
>
> **R3.2**: We believe the results in Table 3 are reasonable. Previous works [1][2] have demonstrated that SFT tends to memorize/overfit training data and exhibits poor generalization, whereas RL-based optimization is more robust [1][2]. More specifically, Table 3 of [2] shows decreased performance of few-shot detection after SFT on some object categories. This explains why SFT underperforms on most categories when only a few examples are available. Regarding training details, as stated in Section 3.1 (Model and Training Details), both RL and SFT models were trained for 30 epochs, with early stopping applied when the reward (for RL) or the loss (for SFT) converged. Thus, both training procedures were run to convergence, ensuring a fair and meaningful comparison.
>
>   [1] Chu et al. Sft memorizes, rl generalizes: A comparative study of foundation model post-training. arXiv preprint arXiv:2501.17161 (2025).
>
>   [2] Liu et al. Visual-RFT: Visual Reinforcement Fine-Tuning. ICCV 2025.
>
> **C3.3 Full-data SFT result.**
>
> **R3.3**: We would like to clarify that the full-data SFT result is already reported in Figure 3. Both GRPO and SFT converge toward the full-data SFT result of 62.91%. This figure shows that few-shot fine-tuned models can achieve performance comparable to full-data SFT at around 200 shots, while also demonstrating GRPO’s strong sample efficiency in the low-shot regime and similar asymptotic behavior as the number of shots increases.
>
> **C3.4 Questions about number of samples.**
>
> **R3.4**: In this work, we follow traditional few-shot learning settings (such as Prototypical Networks [3] and TFA [4]) where shot typically refers to the number of samples per class. There are 26 object categories in VRSBench dataset. Selecting one sample from each class results in 26 samples.
>
>   [3] Snell et al. Prototypical networks for few-shot learning. NeruIPS, 2017.
>
>   [4] Wang et al. Frustratingly Simple Few-Shot Object Detection. ICML, 2020.

---

> > ### Comment · Reviewer_T8hh · 2025-11-27
> > **Reply**
> >
> > I would first like to thank the authors for their detailed response. I have carefully reviewed the rebuttal.
> >
> > However, I would like to raise a broader concern regarding the **novelty and originality** of this work — specifically, how does it fundamentally differ from the well-established method **Seg-Zero**?
> >
> > In addition, I have a few technical questions about the use of **SAM**:   Is the SAM decoder **kept frozen** during different training stages?

---

> ### Author Response · Authors · 2025-11-27
> **Round2**
>
> We sincerely thank reviewer T8hh27 for acknowledging our response and would like to provide some additional details below.
>
> Round2, **C3.1**: How does it fundamentally differ from the well-established method Seg-Zero?
>
> * **R3.1** Our method differs from Seg-Zero in both **supervision signals** and **underlying assumptions**.
>
>   * **Different supervision signals**
>
>     * Seg-Zero (box / point–level supervision).
>
>       The reward for Seg-Zero is composed of `iou_reward`, `point_l1_reward`, and `box_l1_reward`, which are computed from the predicted vs. ground-truth boxes and points. The SAM module is used as a post-hoc to generate masks from boxs and is **not involved in reward computation or optimization**; it is only used to generate masks. In other words, Seg-Zero is based on **box/point-level supervision**.
>
>     * Geo-R1 (mask-level supervision).
>
>       In Geo-R1, the reward is defined at the **mask level**. The SAM module is directly involved in both reward computation and the optimization process. Geo-R1 is based on mask-level supervision rather than box/point-level supervision. Geo-R1 is directly optimized to generate better predicted mask.
>
>
>   * **Different underlying assumptions**
>
>     * Seg-Zero relies on two strong assumptions:
>
>       1. **"Optimal" keypoints for SAM.**
>
>          The “ground-truth” keypoints are obtained via inscribed circles (“we compute the center points of the two largest inscribe circles within the mask, denoted as P1 and P2”). However, these keypoints are **not guaranteed** to be optimal prompt keypoints for SAM.
>       2. **“Optimal” point + “optimal” bounding box ⇒ optimal mask.**
>
>          Seg-Zero implicitly assumes that an optimal point together with an optimal bounding box will yield an optimal predicted mask from SAM, which is also **not guaranteed** in practice.
>
>     * **Geo-R1 does not rely on the above assumptions.**
>       In Geo-R1, the SAM module is used inside the reward calculation, where the reward is computed at the pixel level based on the predicted mask and the annotated mask. The predicted keypoints and bounding box are **optimized freely** and solely to maximize the mask-level reward (e.g., gIoU) with respect to the annotated mask, without assuming that any specific keypoint/box choice is intrinsically “optimal” for SAM.
>
>
> Round2, **C3.2: Is the SAM decoder kept frozen during different training stages?**
> * **Yes.** In Geo-R1, the SAM decoder is **frozen** in all training stages. It is treated as a deterministic, verifiable function and is not finetuned.

---

> > ### Comment · Reviewer_T8hh · 2025-11-28
> > **Reply**
> >
> > Thank you for the authors' response.
> >
> > Based on the discussion above, I have two further concerns:
> >
> > 1. Compared to Seg-Zero (point, box), Geo-R1 introduces an additional mask head, but this mask is predicted from points/boxes generated by the MLLM, with the SAM decoder being frozen. In essence, this still boils down to optimizing how the MLLM produces better points/boxes, rather than modeling masks directly.
> >
> > 2. Moreover, similar approaches have been explored much earlier and the authors are not the first to propose this idea, e.g., SAM-R1 (May 2025).

---

> > > ### Author Response · Authors · 2025-11-28
> > > **Round 3**
> > >
> > > 1. That's true, but we believe it differs significantly from SegZero in terms of key design choices. We find that relying on key points (especially for centers of the largest inscribed circles) for supervision may not be ideal.
> > >
> > > 2. Sam-R1 is an arXiv preprint that has not yet been formally published; it is concurrent with our work (we verified the MaskGIoU reward in April). Per the [ICLR 2026 Reviewer Guide](https://iclr.cc/Conferences/2026/ReviewerGuide), “authors are not required to compare to papers solely on arXiv.”

---

### Official Review · Reviewer_srr5 · 2025-10-31

**Soundness:** 3
**Presentation:** 2
**Contribution:** 1
**Rating:** 2
**Confidence:** 4

**Summary:**

This paper applies reinforcement learning, namely GRPO, to several different tasks in the field of geographical problem solving. The paper compares the RL approach to SFT and shows that RL significantly outperforms SFT in several tasks.

**Strengths:**

+ The empirical results show a strong improvement over the previous SFT methods.
+ The comprehensive experiments substantially demonstrate the claimed improvement for RL vs SFT.

**Weaknesses:**

+ There is as far as I can tell very little conceptually or methodologically novel in this work. The proposed method is an application of GRPO, an algorithm introduced in previous work. The constructed datasets are, again as far as I can tell, a curated subset of previous datasets such as VRSBench. The main result is that on-policy reinforcement learning approaches outperform SFT (behavioral cloning). This is an extremely well-substantiated result in RL, e.g. [1, 2]. Therefore, while it is interesting to see this result in a particular application domain, in my opinion the contribution is not significant enough to warrant publication at ICLR.

+ The paper is somewhat confusing to read, especially with the authors usage of the 'n-shot' terminology. After a close reading I understand that this is used in a meta-learning/task-adaptation sense, where a few examples are provided of a specific task. This is confusing in the language-model context, where few-shot typically refers to a number of provided examples being given in the context of a model before concluding with a final evaluation example.

+ Related to the above, the paper does not have a particularly clear research question or takeaway, due to changing too many variables at once between the SFT and GRPO setting. In particular, as far as I can tell, the 'SFT' setting (which is not defined precisely in the text) refers to training the model with maximum likelihood on the correct response, provided immediately after the prompt. Meanwhile the RL approach uses chain-of-thought reasoning. In order to substantiate a generalizable difference between SFT and RL settings, the paper should compare either RL with no reasoning (i.e. constrained to give the answer immediately) or compare RL with reasoning to SFT over chain-of-thought rollouts filtered for correctness (i.e. true behavioral cloning).


  [1] Ross, Bagnell, Efficient Reductions for Imitation Learning, AISTATS 2010.

  [2] Morales, Sammut, Learning to Fly by Combining Reinforcement Learning with Behavioural Cloning, ICML 2004.

**Questions:**

+ What are the key features that make the work significant and notable enough to be published at ICLR, given the well-established superiority of on-policy RL compared to SFT (behavioral cloning)?
+ What is the motivation for comparing RL with chain of thought against SFT without chain of thought?
+ What are the results when comparing RL without chain of thought to SFT or RL with chain of thought to SFT on successful demonstrations?

---

> ### Author Response · Authors · 2025-11-21
> **To Assigned Reviewer srr5 (Reviewer 2)**
>
> - **C2.1**: Concerns regarding method novelty.
>
>      - **R2.1**: We believe there is a misunderstanding regarding the core contributions of our work. Our paper does not simply apply GRPO to an existing dataset. Instead, we introduce a **new task setting—few-shot geospatial referring expression understanding**—and develop **task-specific reward designs** tailored to this setting. These contributions were explicitly recognized by other reviewers. For example, Reviewer PFsJ noted that our work “introduces a reasoning-centric reinforcement fine-tuning (RFT) framework for few-shot geospatial referring expression understanding” and “develops task-aligned reward functions (IoU, mAP, MaskGIoU) tailored to spatial grounding, including an innovative non-differentiable ‘BBox + SAM’ pipeline for segmentation”. Reviewer T8hh further emphasized that “the application of the RL paradigm to the few-shot remote sensing scene is a valuable and relevant direction for the field” and that the paper “successfully extends from simpler REC and RES tasks to the more complex GREC and GRES tasks”.
>       - In addition, we do not claim to introduce a new dataset; rather, we define and study a new task—few-shot geospatial referring expression understanding—grounded in remote sensing imagery. This task involves unique spatial reasoning challenges and requires specialized reward design and training strategies, which our work proposes and evaluates.
>
> - **C2.2**: Concerns about presentation, especially the 'n-shot' terminology.
>
>     - **R2.2** We agree that the term “n-shot” can carry slightly different meanings in conventional few-shot learning versus in the LLM community. In traditional few-shot classification or detection, *n* typically refers to the number of samples **per class**, whereas in LLM settings it often refers to the number of image–text example pairs provided in the prompt. To avoid confusion, we explicitly define our usage of n-shot in Section 2.2, where we formalize this new task setting and clarify that *n* denotes the number of samplers per class.

---

> ### Author Response · Authors · 2025-11-21
> **Continue**
>
> **C2.3**: Question about SFT and GRPO comparison.
>
>   - **R2.3**: We respectfully disagree with the reviewer’s characterization that we change too many variables between the SFT and GRPO settings. In fact, both approaches use **the same training data and model configurations**. The only difference lies in the training objective (maximum-likelihood vs. GRPO with task-aligned rewards).
>
>  - To further support a fair comparison, we additionally include results for our RL-based method without chain-of-thought reasoning (i.e., the model predicts the final answer directly). As shown in the table below, *Geo-R1 w/o thinking* performs slightly worse than *Geo-R1 w/ thinking*, but still substantially outperforms the SFT counterpart. This indicates that the performance gains primarily stem from the training paradigm (geometry-aware RL) rather than from the reasoning process itself.
>
>     **Table 1. Ablation on CoT**
>     || REC (Acc@0.5) | OVD (mAP) |
>     |:-:|:-:|:-:|
>     SFT | 38.29 | 30.20
>     Geo-R1 w/ thinking | 50.59 | 34.92
>     Geo-R1 w/o thinking | 48.10 |  32.79
>
>   - A core limitation of SFT arises from the next-token prediction (NTP) formulation, which does not inherently support coordinate regression. In contrast, our RL-based framework directly optimizes geometry-aware, task-specific rewards (e.g., box AP, mask GIoU), enabling the model to learn spatially grounded behavior that is difficult to obtain through SFT alone. This observation is consistent with recent evidence showing that **RL-based training without CoT can significantly outperform SFT** in detection tasks [1].
>
>   - Regarding SFT + CoT: we did not compare against this variant because **no high-quality CoT annotations for REC, OVD, GRES tasks exist in the remote sensing domain**. Constructing such a dataset is non-trivial and beyond the scope of this work, but we agree it is an interesting direction for future research.
>
>     [1] Jiang et al. Detect Anything via Next Point Prediction. arXiv preprint arXiv:2510.12798, 2025.
>
>
> **C2.4**: Key contributions of the paper.
>
>   - **R2.4** We would like to emphasize the key contributions:
>
>   1. A new problem setting: We introduce the task of few-shot geospatial referring expression understanding, which has not been previously formulated or addressed in the literature. This task poses unique spatial reasoning challenges that differ fundamentally from standard referring expression tasks in natural images.
>
>   2. A reasoning-centric reinforcement fine-tuning framework: We develop a GRPO-based reinforcement fine-tuning (RFT) framework specifically designed for spatial grounding in remote sensing, enabling the model to reason over both linguistic and geometric cues.
>
>   3. Task-aligned, geometry-aware reward functions: We design reward functions that directly optimize spatial objectives—IoU, mAP, and MaskGIoU—including an innovative non-differentiable “BBox + SAM” pipeline for segmentation reward computation. This allows the model to learn spatially grounded behavior that SFT cannot capture due to its next-token-prediction limitation.
>
>    These contributions go beyond simply demonstrating that on-policy RL outperforms SFT. Rather, they provide a task formulation, a specialized training paradigm, and geometric reward mechanisms that collectively enable effective few-shot spatial grounding in remote sensing—a domain where such capabilities were previously missing.

---

> > ### Author Response · Authors · 2025-11-21
> > **Continue**
> >
> > - **C2.5**: What is the motivation for comparing RL with chain of thought against SFT without chain of thought?
> >    - **R2.5** We agree that it would be more fair to compare RL w/ thinking with SFT w/ thinking. But there's no existing high-quality CoT data in remote sensing domain to enable us training SFT with CoT. We therefore choose to compare SFT to RL w/ and w/o thinking.
> >
> > - **C2.6**: What are the results when comparing RL without chain of thought to SFT or RL with chain of thought to SFT on successful demonstrations?
> >
> >     - **R2.6** - We provide RFT results without thinking in the following. Overall, RL training with thinking provides slightly better performance and much better interpretability. Moreover, RFT consistently outperforms SFT regardless of whether thinking is applied.
> >
> >       || REC (Acc@0.5) | OVD (mAP) |
> >       |:-:|:-:|:-:|
> >       SFT | 38.29 | 30.20
> >       Geo-R1 w/ thinking | 50.59 | 34.92
> >       Geo-R1 w/o thinking | 48.10 |  32.79

---

> > > ### Comment · Reviewer_srr5 · 2025-11-24
> > > **reply**
> > >
> > > Thanks for your reply. I'll address your comments below.
> > >
> > > 1. I don't believe it's reasonable to say that the paper introduces a 'new task setting' with any degree of novelty. The proposed 'referring expression understanding' is just a catch-all term which encompasses the already well-established tasks of referring expression comprehension and referring expression segmentation. There is no unification of the tasks in the paper--they are trained with separate rewards etc. The idea of few-shot or zero-shot REC and RES is also very well-established, e.g. [1] as a random example I found (note it is few-shot since they have some in-context learning examples)
> > > 2. The 'task-specific reward designs' are well-established metrics such as mAP, IoU, etc. There is some reward shaping for the GREC task which is novel. The MaskGIoU metric seems more novel. Overall the paper is significantly over-claiming the contributions. For example the claim of designing a 'GRPO-based reinforcement fine-tuning (RFT) framework specifically designed for spatial grounding in remote sensing' is based on using a very well-known metric for remote sensing (IoU), and otherwise making no algorithmic, architectural, or implementation changes.
> > > 3. SFT comparison. Thanks for giving the comparison with and without thinking for RL training. I think it's very interesting how the score is significantly increased for RL training over SFT, even with no thinking.
> > >
> > > I am sorry to be critical towards the paper. To be clear, I think that the results in this paper would be of interest to many in the geospatial AI field, are of potentially substantial practical interest, and I would expect it to get a good reception at a relevant conference, journal or workshop. However, I don't believe that the contributions are sufficiently novel in the ML sense to recommend featuring it at ICLR.
> > >
> > > [1] https://arxiv.org/pdf/2311.17048

---

> > > > ### Author Response · Authors · 2025-11-26
> > > >
> > > > We sincerely appreciate the reviewer’s follow-up comments. We address the raised concerns below.
> > > >
> > > > 1. On the novelty of the task setting
> > > > * We respectfully disagree with the characterization that our paper “significantly over-claims” its contributions. We fully acknowledge that referring expression comprehension (REC) and referring expression segmentation (RES) are well-established tasks, and that few-shot or zero-shot variants have been explored in natural-image domains.
> > > > * However, our contribution is not a generic claim of inventing REC/RES; rather, it lies in introducing and formalizing few-shot geospatial referring expression understanding in aerial imagery. Thus, the novelty we claim is domain-specific and task-specific, not a rebranding of existing concepts.
> > > >
> > > > 2. On task-specific reward design
> > > > * We agree with the reviewer that mAP and IoU are well-known metrics. Our claim is not that these metrics are new, but that:
> > > >     * Integrating them into a reinforcement fine-tuning framework for spatial grounding in aerial imagery is new,
> > > >     * The MaskGIoU reward and the BBox+SAM pipeline for GRES are novel contributions tailored to the unique needs of referring segmentation in remote sensing images.
> > > > * Our contribution is therefore not the metrics themselves, but the geometry-aware reward formulation, which enables RFT to optimize spatial grounding behavior that cannot be learned through next-token-prediction objectives.
> > > >
> > > > 3. On the RL vs. SFT comparison
> > > > * We appreciate the reviewer’s acknowledgment that RL, even without chain-of-thought, substantially outperforms SFT. As mentioned in our previous response, this behavior is expected:
> > > >     * SFT is constrained by next-token-prediction and is not designed for coordinate regression.
> > > >     * Our RL framework directly optimizes task-aligned spatial rewards (box AP, MaskGIoU), enabling learning signals that are otherwise inaccessible.
> > > > * This phenomenon is also consistent with recent findings in detection-oriented LLM training, e.g.,Jiang et al., “Detect Anything via Next Point Prediction”, arXiv 2025, which reports similar patterns.
> > > >
> > > >
> > > > 4. On the appropriateness of ICLR as a venue
> > > > * We appreciate the reviewer’s perspective and fully understand the distinction between fundamental ML and more application-driven ML. However, we would like to clarify our rationale:
> > > >     * ML research includes both algorithmic advances and high-impact domain applications, and both are essential to expanding the practical reach of machine learning. Many ICLR/NeurIPS publications focus on domain-specific adaptation of ML techniques (e.g., medical imaging, molecular modeling, robotics, remote sensing).
> > > >     * Our submission selected the primary area “applications to computer vision, audio, language, and other modalities”, which we believe accurately reflects the scope of our paper and aligns with ICLR’s stated areas of interest.
> > > >     * Other reviewers also emphasized that applying RL-based fine-tuning to few-shot geospatial reasoning represents a meaningful step forward for real-world ML deployment.
> > > > * We therefore respectfully maintain that the work fits within ICLR’s scope, particularly under application-driven ML.
> > > > &nbsp;
> > > > * Publications focus on domain-specific adaptation of ML techniques
> > > >
> > > >     * SAMRS: Scaling-up Remote Sensing Segmentation Dataset with Segment Anything Model (Wang et al. Neurips2023)
> > > >       - https://neurips.cc/virtual/2023/poster/73494
> > > >
> > > >     * Cross-Scale MAE: A Tale of Multiscale Exploitation in Remote Sensing (Tang et al. Neurips2023)
> > > >       - https://neurips.cc/virtual/2023/poster/72787
> > > >
> > > >     * Remote Sensing Vision-Language Foundation Models without Annotations via Ground Remote Alignment (Mall et al. ICLR2024)
> > > >       - https://iclr.cc/virtual/2024/poster/17503
> > > >
> > > >     * MMed-RAG: Versatile Multimodal RAG System for Medical Vision Language Models (Xia et al. ICLR2025)
> > > >       - https://iclr.cc/virtual/2025/poster/28145
> > > >
> > > >     * TEOChat: A Large Vision-Language Assistant for Temporal Earth Observation Data (Irvin et al. ICLR2025)
> > > >       - https://iclr.cc/virtual/2025/poster/28289

---

### Official Review · Reviewer_PFsJ · 2025-11-01

**Soundness:** 2
**Presentation:** 3
**Contribution:** 2
**Rating:** 6
**Confidence:** 4

**Summary:**

This paper proposes Geo-R1, a reasoning-centric reinforcement fine-tuning (RFT) framework for few-shot geospatial referring expression understanding (REU). Instead of traditional supervised fine-tuning, Geo-R1 trains vision-language models to “reason first, then act,” generating explicit <think> reasoning chains before outputting bounding boxes or masks. Using Group Relative Policy Optimization (GRPO) and task-aligned rewards (IoU, mAP, MaskGIoU), the method directly optimizes for localization accuracy even in data-scarce regimes. Evaluated on three few-shot benchmarks (VRSBench-FS, NWPU-FS, EarthReason-FS), the results show that Geo-R1 consistently outperforms SFT by 5–15 points and generalizes better across the datasets considered. The authors also present analyses to demonstrate sample-efficiency, training stability, and interpretability (through reasoning traces) for the approach.

**Strengths:**

Originality:
- Introduces a reasoning-centric reinforcement fine-tuning (RFT) framework for few-shot geospatial referring expression understanding, a novel combination of reasoning-driven RL and remote-sensing multimodal grounding.
- Develops task-aligned reward functions (IoU, mAP, MaskGIoU) tailored to spatial grounding, including an innovative non-differentiable “BBox + SAM” pipeline for segmentation.

Quality:
- Empirical results are consistent, and reproducible across three few-shot benchmarks (VRSBench-FS, NWPU-FS, EarthReason-FS), each designed for distinct sub-tasks (REC, OVD, GRES).
- Decent methodological rigor: identical base models, equalized training budgets, and clear comparisons between SFT and GRPO.

Clarity:
- The paper is well-structured, with clear motivation, and intuitive figures (especially Fig. 1 pipeline diagram and learning-curve plots).
- The reasoning format think…think → answer…answer makes the method interpretable and easy to follow, and technical choices (rewards, GRPO details) are described transparently.

Significance:
- Demonstrates a sample-efficient alternative to SFT for adapting vision-language models in data-scarce geospatial settings.
- Establishes reasoning-based RL as viable for improving few-shot multimodal generalization.
- By releasing the few-shot datasets and code, the work provides a basis for future research on interpretable, reasoning-enhanced VLMs for remote sensing.

**Weaknesses:**

- Marginal methodological novelty:
The paper has a modest conceptual leap. The “reason-first, then-act” framing builds directly on recent reasoning-centric reinforcement learning trends (e.g., DeepSeek-R1, o1), extending them to geospatial referring tasks rather than introducing new algorithmic principles. The innovation lies in task-specific rewards and adaptation to remote sensing rather than in a new learning paradigm.
- Insufficient ablation and causal transparency:
The study does not clearly isolate which design components (reasoning format enforcement, task-aligned reward metrics, or GRPO optimization) drive the reported gains. Without systematic ablations or controlled reward variations, it is difficult to assess whether improvements stem from the reasoning structure itself or from auxiliary factors such as reward shaping or hyperparameter tuning.
- Lack of uncertainty quantification and robustness analysis:
Results are presented without standard deviations, confidence intervals, or variance across random seeds. In a few-shot context where performance can fluctuate substantially, the absence of uncertainty reporting undermines claims of reliability and generality. A simple table showing variance across multiple runs would materially strengthen the paper’s empirical credibility.
- Asymmetric comparison in segmentation (GRES):
Because the reward pipeline is non-differentiable, the GRES experiments exclude a direct SFT baseline and instead compare against another RL-based system (SegEarth-R1). This asymmetry limits interpretation: the benefit of reinforcement fine-tuning from the design advantage of the BBox + SAM mechanism cannot easily be separated.
- Compute intensity relative to problem scale:
Although Geo-R1 is positioned as a data-efficient adaptation method, training requires roughly 8×H100 GPUs for 10–20 hours. Such requirements limit accessibility and partially offset the appeal of few-shot efficiency.
- Narrow evaluation scope:
All benchmarks use RGB aerial imagery drawn from related distributions. The approach’s robustness to domain shifts remains untested. As a result, the claimed cross-dataset generalization speaks more to benchmark transfer than to real-world generalization across modalities.
- Missing competitive adaptation baselines:
The comparison set omits other modern low-data adaptation strategies such as LoRA, adapter tuning, or visual prompt tuning, which are computationally lighter to reinforcement fine-tuning. Including at least one of these would better contextualize Geo-R1’s advantages.

**Questions:**

Questions and Rebuttal Suggestions for the Authors
- Ablations on reward design:
Could you provide quantitative ablations isolating the effects of each reward component: (i) format reward enforcing structure, (ii) task-aligned metric rewards (IoU, mAP, MaskGIoU), and (iii) any length or penalty terms? Understanding which components are most responsible for the observed gains would clarify the causal mechanism behind Geo-R1’s improvements.
- Variance and robustness reporting:
Please include results from multiple random seeds (or at least three runs) with standard deviations or confidence intervals. This is essential for assessing robustness, especially in few-shot regimes where noise and initialization effects can be substantial.
- Cross-task generalization and domain shifts:
How does Geo-R1 perform under genuine domain shifts (different sensors, seasons, or geographies)? Even one experiment using multispectral or SAR imagery, or data from a distinct geographic region, would substantively strengthen claims of generalization.
- GRES comparison and fairness:
Since SFT cannot be directly applied to the non-differentiable BBox + SAM pipeline, could you comment on whether a proxy SFT baseline (e.g., trained with a differentiable mask loss) was attempted? Clarifying this would help attribute the segmentation gains to RL rather than to pipeline differences.
- Compute and scaling trade-offs:
Could you quantify Geo-R1’s compute cost relative to SFT and discuss whether smaller or more efficient variants (e.g., lower-precision training, distillation, or smaller backbones) preserve most of the gains? This would help judge its practicality.
- Comparison to lightweight adaptation methods:
Were LoRA, adapter tuning, or visual prompt-based few-shot methods considered? Even a brief empirical or conceptual comparison would better position Geo-R1 within the broader adaptation landscape.

---

> ### Author Response · Authors · 2025-11-21
> **To Assigned Reviewer PFsJ (Reviewer 1)**
>
> - **C1.1 Ablations on reward design**:
>
>     - **R1.1**: Thank you for the suggestion. We would like to clarify that only the format reward is shared across all tasks, while the task-aligned rewards (IoU, mAP, MaskGIoU) are task-specific and therefore not applied jointly. As a result, an ablation isolating each component within a single task is not directly applicable. In the following, we provide quantitative ablations showing the contribution of the format reward, as well as a task-specific length penalty reward for the  OVD task, which helps clarify the role of each component in driving Geo-R1’s improvements.
>
>         - **Table 1. Ablation on Format Reward**
>
>             ||REC (Acc@0.5)|GRES-val (gIoU)|GRES-test (gIoU)|OVD(mAP)
>             |:-:|:-:|:-:|:-:|:-:|
>             |Geo-R1 w/ format reward|50.59|57.78|58.27|34.92|
>             |Geo-R1 w/o format reward|48.23|56.61|57.24|34.64|
>
>         - Table 1 shows that without the format reward, Geo-R1 exhibits slightly degraded performance on both REC and GRES tasks. Empirical results also show that Geo-R1 w/o format reward requires much longer training time to converge (about 1.6× longer for REC, 1.7x longer for OVD and 1.3x longer for GRES) compared to Geo-R1 w/ format reward.
>
>         - **Table 2. Ablation on OVD Length Penalty**
>
>             ||OVD (mAP)|
>             |:-:|:-:|
>             |Geo-R1 w/ length penalty|34.92|
>             |Geo-R1 w/o length penalty|27.52|
>
>         - Table 2 shows that without the length penalty reward, Geo-R1 exhibits significantly degraded performance on the OVD task.
>
>
> - **C1.2: Variance and robustness reporting**:
>
>     - **R1.2**: Thank you for this valuable suggestion. We have re-sampled the training data using 10 different random seeds and re-run the experiments for the REC and GRES tasks. For OVD, we re-sampled with 3 different random seeds due to time constraint and report 5-shot result (10-shot setting for OVD is hard to re-sample different samples). As shown in the table below, our model exhibits stable performance across different seeds, with standard deviations below 1%.
>
>       | Task |Metrics| Result from the Paper (Seed 42)| Average +- std|Seed 43 | Seed 44 | Seed 45 | Seed 46 | Seed 47 | Seed 48 | Seed 49 | Seed 50 | Seed 51 | Seed 52 |
>       | :-: | :-: | :-: | :-: |:-: | :-: | :-: | :-: | :-: | :-: | :-: | :-: | :-: | :-: |
>       |REC|Acc@0.5|50.59|49.46 $\pm$ 0.95|49.90|49.76|50.34|49.77|50.28|47.43|	49.20|	49.61|	50.33|	47.99|
>       |GRES-Val|gIoU|57.78|59.19 $\pm$ 0.31|59.08	|58.78	|59.41|	59.58|	59.57	|58.80	|59.33|59.52|58.89|58.94|
>       |GRES-Test|gIoU|58.27|58.05 $\pm$ 0.63|59.00|	58.20|	58.18|	57.27	|58.41|57.10|58.47|58.59|58.17|	57.11|
>       |OVD|mAP|33.20|33.68 $\pm$ 0.97|34.36|33.71|34.79|33.50|32.10|34.20|34.74|32.09|32.87|34.48|
>
>
> - **C1.3: Cross-task generalization and domain shifts**:
>
>     - **R1.3**: Thank you for raising this important point. We fully agree that robustness under domain shift is a critical aspect when developing EO foundation models. In this work, we did not evaluate Geo-R1 under genuine cross-modal domain shifts (e.g., Optical → SAR/IR) because Geo-R1 is not intended to serve as a comprehensive remote sensing foundation model. Instead, our aim is to propose an efficient few-shot adaptation strategy built on top of existing foundation models. To demonstrate the generalization capability of Geo-R1, we already evaluated the performance of Geo-R1 under cross-dataset setting in Section 4.2.
>      - Regarding cross-sensor (modality) generalization, we expect Geo-R1’s performance on SAR/IR imagery to be limited, primarily because the underlying base model (Qwen2.5-VL) lacks pre-training on these modalities. Nonetheless, the proposed method is model-agnostic and could naturally extend to SAR/IR data when paired with a base VLM that is trained on such modalities.
>      - With respect to cross-season and cross-region generalization, our literature review did not identify any well-curated vision–language datasets that provide explicit season or region annotations suitable for such evaluation. If the reviewers are aware of such benchmarks, we would be very happy to incorporate them into our experiments.

---

> ### Author Response · Authors · 2025-11-21
> **Continue**
>
> - **C1.4: GRES comparison and fairness**
>
>     - **R1.4**: You are correct that standard SFT for MLLMs does not support optimization with a differentiable mask loss. A common “proxy SFT” strategy is to introduce a dedicated segmentation token along with a trainable decoder, allowing the model to learn a pixel-level mask through a differentiable mask loss. This is exactly the architecture employed by SegEarth-R1 (see Section 3.2.2). Accordingly, we chose SegEarth-R1 as a strong SFT-based baseline to ensure a fair comparison and to demonstrate that our GRPO-based pipeline outperforms the conventional SFT + decoder approach.
>
>
> - **C1.5: Compute and scaling trade-offs**
>
>     - **R1.5**: In our experiments, the training cost of GRPO is approximately 5–6 times higher than that of SFT. This difference is primarily due to two factors: (1) the longer chain-of-thought reasoning process compared to directly generating final results, and (2) the multiple rollouts required to compute the group advantages for policy optimization. Despite this increase, the total training time remains manageable (typically less than one day). Recent efficient RL training libraries, such as vLLM [1], verl [2], can also be used to improve training efficiency of our model. More importantly, the inference cost of our RL-based model remains comparable to that of SFT.
>     - Furthermore, our ablation studies on model size (Figure 4) show that a small model (Geo-R1-3B) preserves most of the performance gain. It's technically reasonable to apply lower-precision training or model distillation to further enhance the model, but we did not apply these techniques since they usually have a minor benefit to model performance (compared to the performance gain from RL) and are usually considered when GPU memory is not enough. We expect these techniques would be useful for resource-limited scenarios, e.g., in-orbit object detection.
>     - Finally, we would like to point out that Geo-R1 is designed to address the challenges of high annotation costs and data scarcity (few-shot) in the remote sensing domain. As we prove in our paper, GRPO yields significant improvements in data efficiency compared to SFT. The increased compute cost is thus a worthwhile trade-off for the substantial reduction in data and annotation cost.
>     - [1] Kwon et al. Efficient memory management for large language model serving with pagedattention. Proceedings of the 29th symposium on operating systems principles. 2023.
>     - [2] Sheng et al. Hybridflow: A flexible and efficient rlhf framework. Proceedings of the Twentieth European Conference on Computer Systems. 2025.

---

> ### Author Response · Authors · 2025-11-21
> **Continue**
>
> **C1.6: Comparison to lightweight adaptation methods: Were LoRA, adapter tuning, or visual prompt-based few-shot methods considered?**
>
>   - **R1.6**: We acknowledge that lightweight adaptation methods—such as LoRA, adapter tuning, and visual prompt tuning—have been widely adopted for few-shot adaptation of VLMs for classification tasks. However, these approaches are not designed to address the core challenges of our referring expression understanding tasks, which require generating bounding boxes or segmentation masks.
>
>   - A key limitation of SFT-based methods in this context stems from the next-token-prediction (NTP) nature of LLMs, which does not inherently support coordinate regression. Consequently, lightweight adaptation techniques—despite their parameter efficiency—remain within the SFT regime and inherit the same training-objective constraints. In contrast, our RL-based training framework enables geometry-aware optimization through task-specific rewards (e.g., box AP or mask GIoU). This represents a shift in the adaptation paradigm: *rather than solely improving parameter efficiency, our method directly optimizes training objectives that SFT-based lightweight adapters cannot effectively handle*.
>
>    - Finally, we also note that lightweight adaptation methods could be used together with our RL-based training framework. While it is feasible to apply LoRA in our method, prior work has shown that such parameter-efficient approaches often yield limited performance improvements compared to full-model tuning in MLLMs, and are typically adopted when GPU memory is a critical constraint. This is consistent with observations from [1][2], which note that LoRA primarily targets memory efficiency rather than achieving superior performance. To further validate this, we train our Geo-R1 model using LoRA (rank = 64, alpha = 128), and the results are shown in the Table below. Results show that combining LoRA with Geo-R1 slightly hurts the performance and does not save much training time compared with full fine-tuning (9 hours vs. 10 hours). This is because the rollout operation dominates the GRPO training process, while updating the LLM accounts for only a small portion of the total time.
>
>         ||REC (Acc@0.5)|GRES-val (gIoU)|GRES-test (gIoU)|
>         |:-:|:-:|:-:|:-:|
>         |Geo-R1 LoRA|47.62|51.46|53.89|
>         |Geo-R1 full-finetune|50.59|57.78|58.27|
>
>       [1] Hu et al. Lora: Low-rank adaptation of large language models. ICLR, 2022.
>
>       [2] Shen et al. Vlm-r1: A stable and generalizable r1-style large vision-language model. arXiv preprint arXiv:2504.07615 (2025).

---

### Author Response · Authors · 2025-11-21
**Response to All Reviewer**

We sincerely thank all reviewers for their time and insightful comments. Please find the responses (**R**) to specific comments (**C**).

We also updated the PDF according to reviewers' suggestions.

---

### Comment · Area_Chair_nbZz · 2025-11-27
**Please check the rebuttal**

Dear Reviewers,

The authors have posted their rebuttal. Could you please take a moment to review their responses and check whether your concerns have been adequately addressed? If possible, kindly initiate the discussion at your earliest convenience.

Your timely assistance is essential for keeping the review process on track. Thank you very much for your support and contribution.

Best regards,
Your AC

---

### Note · Authors · 2025-11-29

I have read and agree with the venue's withdrawal policy on behalf of myself and my co-authors.